# ZeroTS: Zero-shot time series forecasting via multi-party data-model interaction

## Abstract

Time series forecasting (TSF) is a fundamental task in artificial intelligence, with applications ranging from weather prediction, stock market analysis to electricity demand forecasting. While existing models, particularly large language models (LLMs) tailored for TSF, primarily focus on improving accuracy and generalization through pre-training and fine-tuning, zero-shot prediction without task-specific fine-tuning, still remains underexplored. This limitation arises from the restricted scalability and flexibility of current LLMs for TSF, which struggle to fully capture the interactions between data and model. In this work, we introduce ZeroTS, a novel approach that bridges open-world knowledge with inherent data regularities by constructing multi-party interactions between data and models. On the data side, we propose a TS-RAG (Retrieval-Augmented Generation for Time Series), which efficiently retrieves both meta and series information, enabling diverse domain-specific time series to be used as prompts. On the model side, we develop a reinforcement learning framework that treats ground-truth as environments, providing error feedback to optimize a smaller model and harnessing the capabilities of LLMs. This allows ZeroTS to incrementally approach inherent data regularities while iteratively refining its outputs. We validate ZeroTS via extensive experiments on zero-shot and long-horizon forecasting. ZeroTS achieves best or second best results with comparative parameters, 1/4 memory and 1/7 inference speed, demonstrating its efficiency and effectiveness. Our results highlight the potential of Data-LLM interactions for zero-shot learning with acceptable parameters, opening new avenues on research of this underexplored area.

## 1 Introduction

Time series forecasting (TSF) is one of the fundamental capacities for data-driven artificial intelligence, which facilitates the progress of various urban and scientific applications such as weather forecasting (Jin et al., 2023), stock prediction (Singh & Srivastava, 2017) and electricity planning (Reddy et al., 2023; Contreras et al., 2003). Zero-shot prediction, an emerging technique for object detection, enables learning scheme free of repeated fine-tuning and usually output the data point as a new class or derive a description based on learning with historical data (Bansal et al., 2018; Li et al., 2019). In the field of time-series forecasting, zero-shot time series prediction is also seriously required in data-scarce scenarios such as new cities, extreme weather conditions and cross-domain adaptation. However, due to the large-scale patterns and trends in real-world series, it is a challenging and under-explored task.

Current efforts on time series forecasting can be categorized into traditional model like moving average-based ARIMA (Contreras et al., 2003), recurrent-based LSTM (Yu et al., 2019), convolution-based Temporal Convolution Network (TCN) (Hewage et al., 2020), and Transformer-based methods (Zerveas et al., 2021), where they can mostly well address the specific single task in single domain. With the rising of Large Language Models (LLM), researchers start to exploit the open-world knowledge from LLM to help time series prediction. For the first time, Time-LLM transforms numerical series into characters and realizes prediction with LLM backbones (Zhou et al., 2023b). Even so, general LLM solutions solely rely on parameterized textualized knowledge and fail to filtrate specific and conditional patterns for achieving more accurate results, which is so-called *hallucination* in LLM. To this end, we can summarize two issues that limit existing LLMs to implement zero-shot prediction. *1) Insufficient scalability and flexibility.* With increasing pro-

duction of time series in open world, parameterized model usually fails to dynamically incorporate the increasingly real-time data into predictions. And most LLM solutions without any learnable parameters cannot adapt themselves to new target data (Zhou et al., 2023b; Gruver et al., 2024). *2) Lacking data-model interactions.* Existing methods usually fail to provide feedback of how LLM match the task, i.e., how to evaluate the quality of results from LLM and adjust LLM prompts for achieving better results. Thus, interactions of coupling inherent knowledge within structural data and textualized knowledge in LLM are significantly neglected in an unified training pipeline.

Actually, constructing multi-party interactions between LLM and original structural data to iteratively improve the coherence between large model, small model and targeted data, can activate the power of LLM for achieving zero-shot time-series prediction. To this end, we propose our ZeroTS from both data aspect and model aspect as follows. As illustrated in Figure. 1, from data aspect, data from different domains can potentially share similar evolution patterns. The more diverse and richer time-series, the more patterns

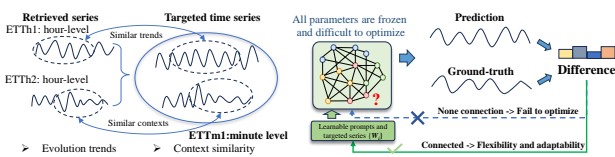

Figure 1: Motivation. (a) Series from different domains share similar patterns, where they may experience similar context (e.g., peak followed by fluctuations). (b) Compared to none feedback for optimization, involving feedback can connect the model output with arrived new time-series for model optimization, improving flexibility and adaptability.

are available. Fortunately, RAG integrates capacity of information retrieval with generative ability of LLM, enabling the prediction system to automatically pick up the key samples of interest from the database and provide potential cues to suppress the hallucinations and deviations of the model (Fan et al., 2024; Peng et al., 2024). On model aspect, most LLM solutions either directly take the output of LLM as final results (Zhou et al., 2023b; Gruver et al., 2024) thus lacking the specified optimization, or explore an Low-Ranking Adaptation (LoRA) to fine-tune partial layers of LLM (Hu et al., 2021; Zhou et al., 2023a) thus leading to excessive computational loads. In contrast, designing a lightweight learnable module to couple the feedback of LLM and inherent data patterns can enable the learnable model to fully assimilate both textualized knowledge from large model and real-world data regularities, further improving the quality of prediction.

Even so, coupling the RAG with time series is an emerging topic that has never been reported in previous literature. Given properties of time series, i.e., informative meta values, multi-grained pattern and complex trends, the coupling is still faced with following challenges. **First**, How to construct time series-oriented **database (dataset)** for supporting effective and efficient retrieval augmented generation? **Second**, how to optimize the dual pipeline to improve quality of LLM output, i.e., optimizing the representation of targeted time-series and constructing the informative and helpful prompts from retrieved time series to help better implement zero-shot time series prediction? **Thirdly**, how to construct the lightweight module, to not only assimilate the intelligence from data side, but also simultaneously exploit knowledge from LLM to cooperatively realize feedback-based optimization for achieving high-quality prediction results?

To address above challenges in zero-shot time-series prediction, we propose a novel time-series learning framework ZeroTS, which leverages the data and model to cooperatively realize interactions and enables a lightweight parameter-efficient model to activate the power of LLM. The technical contributions are three-fold. 1) We first introduce RAG into zero-shot time-series prediction, devise a meta-value structure to accommodate informativeness of various series, and a modified quick retrieval HSNSW with hybrid affinity measurement for efficient retrieval. 2) We propose a reinforcement scheme to assimilate knowledge from LLM, and interact LLM with ground-truth as feedback to optimize the coherence between RAG and LLM, so that it can constantly absorb the law of real data, and guide the output of LLM closer to the facts. 3) We evaluate our ZeroTS over both zero-shot and long-term prediction settings. It achieves satisfactory performances, improving at most 6% and obtaining almost best or second best results with limited memory and quickest test time, i.e., 1/4 memory and 1/7 speed against other parallel models.

## 2    RELATED WORK

**Time-series forecasting.** Time Series Forecasting has evolved significantly from traditional statistical approaches to modern machine learning techniques. Classical methods like Autoregressive Integrated Moving Average Model (ARIMA) and Exponential Smoothing (ETS) provided a foundational framework by modeling linear relationships within time series (Box et al., 2015). However, these methods often struggled with capturing non-linear dynamics in complex systems. Machine learning algorithms such as decision trees and support vector machines later improved performance by addressing non-linearities (Hyndman, 2018). Deep learning techniques like Long-Short Term Memory networks (LSTM) (Graves & Graves, 2012) and Temporal Convolutional Networks (TCNs) (Bai et al., 2018) has emerged, excelling in identifying complex temporal dependencies and long-range patterns within sequential data. Besides, to advance the multi-variate series forecasting, Graph neural network (GNN)-based series forecasting are proposed, such as MTGNN (Wu et al., 2020; Zhou et al., 2023b) adaptively captures variable-wise relations, and CrossGNN dynamically captures scale-level and variable-level correlations (Huang et al., 2023). These models have been widely adopted for various tasks such as finance, weather prediction, and traffic demand.

**LLM4TS and RAG.** LLM4TS (Large Language Models for Time Series) represents a novel approach that leverages the capabilities of large language models for time series forecasting. Unlike traditional methods, LLM4TS utilizes self-attention mechanisms to capture global patterns and dependencies in sequential data, providing enhanced performance over longer time horizons (Chang et al., 2023). Recent advancements have shown that large models, such as Transformers, are highly effective in understanding the temporal structure of time series, enabling them to generate more accurate forecasts (Vaswani, 2017). For example, FPT (Zhou et al., 2023a) fine-tunes a BERT encoder to perform time series forecasting. Similarly, Zhang et al, (Zhang et al., 2023) introduce Meta-Transformer, a framework for finetuning a language model for non-text modalities, including time series. Huang, et, al (Huang et al., 2024) devises a LeRet to integrate language knowledge with structural data to realize accurate LLM-empowered forecasting.

Retrieval-Augmented Generation (RAG) enhances large language models by allowing them to retrieve external information dynamically. This technique combines the strengths of generative models, such as GPT, with retrieval mechanisms that automatically retrieves relevant knowledge from external databases (Lewis et al., 2020). By incorporating external data into the forecasting process, RAG improves the model's ability to make informed predictions, especially when facing scenarios with limited historical data or requiring domain-specific knowledge (Karpukhin et al., 2020).

The combination of LLM4TS and RAG can offer significant advantages. LLM is equipped with great open-world knowledge in parameters for forecasters. RAG enhances LLM4TS's forecasting by incorporating relevant external knowledge, like economic indicators or weather series, leading to better contextual understanding and more precise predictions in specific domains. Designing such hybrid model can potentially tackle the data sparsity and uncertainty by filling missing information with external knowledge. But unfortunately, there is very limited works investigating such new learning scheme, i.e., LLM4TS with RAG.

**Zero-Shot Forecasting.** Zero-Shot Forecasting has emerged as an exciting area in time series forecasting, offering the ability to make predictions where task-specific training data is either unavailable or extremely limited (Bansal et al., 2018; Li et al., 2019). Zero-shot forecasting utilizes pre-trained models that can generalize across domains, allowing them to make accurate predictions in novel contexts without retraining. This capability is particularly useful in cases where collecting labeled time series data is costly or impractical. LLMTime (Gruver et al., 2024) explored the potential of zero-shot forecasting by demonstrating how pre-trained language models can be applied to time series data, generating reliable forecasts without the need for extensive fine-tuning. The combination of LLM4TS and zero-shot forecasting allows for the flexible application of models to new domains, greatly enhancing the practical utility of these systems in real-world forecasting tasks. However, given the exciting recent research, how to incorporate the RAG bonus into LLM to enable efficient and effective zero-shot forecasting still remains challenging and unclear.

## 3    PRELIMINARIES AND PROBLEM DEFINITION

**Time-series forecasting (TSF).** Given time series $\{x_1, x_2, ..., x_t, ...\}$, and a well-trained foundation time-series forecasting model $\mathcal{M}$, the goal of time series forecasting is to predict the following

consecutive $Q$ steps by optimizing all learnable parameters $\Theta$ in $\mathcal{M}$, i.e., $\widehat{Y} = \mathcal{M}(X; \Theta)$, where $\widehat{Y} = \{\widehat{y}_{T+1}, \widehat{y}_{T+2}, ..., \widehat{y}_{T+Q}\}$, $X = \{x_1, x_2, ..., x_T\}$, $\Theta$ represents all learnable parameters $\{W_i\}$.

**Retrieval Augmented Zero-shot TSF.** Given a set of series from different domains $\mathcal{G} = \{\mathcal{G}_1, \mathcal{G}_2, ..., \mathcal{G}_K, ...\}$, where each domain consists of corresponding time-series $\mathcal{G}_k \leftarrow \{X^k | x_1^k, x_2^k, ..., x_T^k\}$. We construct them into an auxiliary **database (dataset)** $\mathcal{D}$ where the domain and series can be dynamically updated and increasing. Given one targeted series $X^T$, we retrieve $K$ most helpful auxiliary series $\mathbb{X}^R = \{X_0^R, X_1^R, ..., X_K^R\} \in \mathcal{D}$, and implement forecasting $\widehat{Y}^T = \mathcal{M}^*(X^T | \mathbb{X}^R)$, where $\mathcal{M}$ is the modified learning pipeline with LLM, $\widehat{Y}^T = \{\widehat{y}_{T+1}, \widehat{y}_{T+2}, ..., \widehat{y}_{T+Q}\}$, $\mathbb{X}^R = \{X_1^R, X_2^R, ..., X_K^R\}$. The condition can be considered as the retrieved augmented series for implementing RAG prompts.

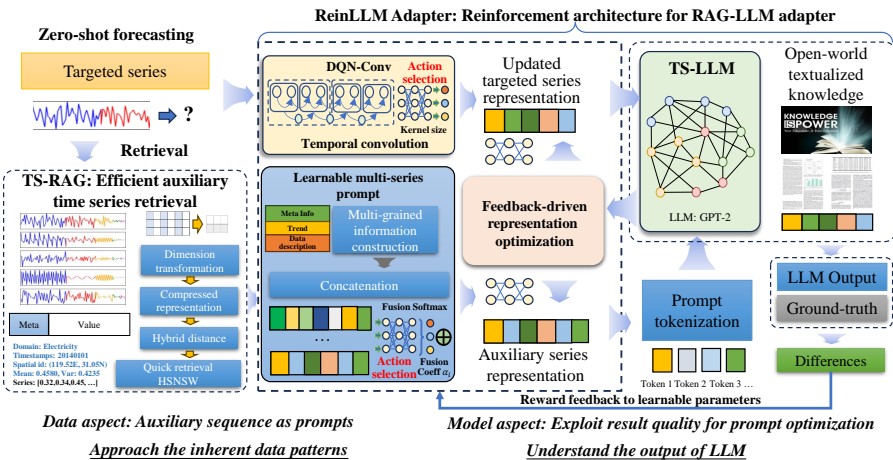

Figure 2: Framework overview of ZeroTS.

# 4 METHODOLOGY

Our ZeroTS addresses the zero-shot time series forecasting, which leverages the power of LLM and coheres the gap between the parameterized knowledge in LLM and structural real-world data via devising a parameter-efficient adapter. Our ZeroTS consists of two major components, a Time-series Retrieval Augmented Generation (TS-RAG) for efficiently retrieving high-quality exogenous series from an auxiliary **database (corresponding to an external dataset)** $\mathcal{D}$, and a ReinLLM, an reinforcement scheme that simultaneously calibrates parameter-efficient adapter and forces LLM to approach the data property. The framework overview is illustrated in Figure. 2. More technical details, theoretical guarantee as well as the efficiency are comprehensively discussed in Sec. A.1 and A.3 of Appendix.

## 4.1 EFFICIENT AUGMENTED TIME SERIES RETRIEVAL

As discussed, auxiliary series can help provide informative hints for guiding prediction of targeted series. To ensure augmented prediction, the first task is to construct a retrieval **dataset** and efficiently retrieve the augmented auxiliary time-series.

**Structural key-value generation for RAG dataset.** Time series is a kind of structural data with various meta information such as domain category, timestamps, and identity of corresponding series. The construction of our RAG database for auxiliary series retrieval can be two-fold, the meta information and deterministic value of series. To standardize the data formats and facilitate the retrieval process, we first generate a structural key-value sequence in RAG database for efficient retrieval. We argue that both textualized meta information and numerical statistics are important in retrieval, thus our keys consist of the meta information and especially the numerical statistics. We select the domain category, timestamps and spatial location or user identification as the meta information, and calculated mean and variance are also considered meta information for similarity-based retrieval.

For the values of numerical series, we normalize them to eliminate the influences of fluctuation ranges of each series in different domain categories. Given the $i$-th time series in training set $\boldsymbol{X}_i$, we update each element into the normalized version by extracting the expectation $\mathrm{avg}(\boldsymbol{X}_i)$ and standard deviation $\mathrm{std}(\boldsymbol{X}_i)$ of corresponding sequence, $x_i' = \frac{x_i - \mathrm{avg}(\boldsymbol{X}_i)}{\mathrm{std}(\boldsymbol{X}_i)}$. For easy notation, we let $x_i \leftarrow x_i'$ update sequences into normalized version.

**Data-driven series retrieval.** As time-series with similarities tend to share common evolution patterns, we explore the RAG series database for augmented learning. To facilitate the retrieval, we devise a compressed representation to maximumly reduce the computation loads and propose a hybrid similarity to extract the optimal sequence for augmentation. Given series $\boldsymbol{X}_0^R \in \mathbb{R}^{1 \times Q}$ in the retrieved **database (dataset)**, the compressed representation can be obtained by minimizing the following reconstruction objective,

$$\min ||\boldsymbol{X}_0^R - (\boldsymbol{X}_0^R * \boldsymbol{W}_c) * \boldsymbol{W}_c'||_2 \tag{1}$$

where the $\boldsymbol{W}_c \in \mathbb{R}^{q \times p}, \boldsymbol{W}_c' \in \mathbb{R}^{p \times q}$, $p = \rho q (0 < \rho < 0.5)$. When the model is well-trained, we adopt the compressed matrix $(\boldsymbol{X}_0^R)_c = \boldsymbol{X}_0^R * \boldsymbol{W}_c$ as the compressed representation and transfer the $\boldsymbol{W}_c$ to any other series for reducing the dimension of series.

Regarding the computation of similarity, we adopt a hybrid similarity metric by incorporating both cosine similarity for trend modeling and Euclidean distances for value discrepancy. Given a targeted sequence $\boldsymbol{X}^T$ and series from auxiliary **database (dataset)** $\boldsymbol{X}_i^R$, we can obtain the similarity by,

$$\mathrm{Sim}(\boldsymbol{X}^T, \boldsymbol{X}_i) = \cos(\boldsymbol{X}^T, \boldsymbol{X}_i) + 1/dist(\boldsymbol{X}^T, \boldsymbol{X}_i) \tag{2}$$

where $\cos(\boldsymbol{X}^T, \boldsymbol{X}_i)$ and $dist(\boldsymbol{X}^T, \boldsymbol{X}_i)$ denote cosine similarity and Euclidean distance. Eq. 2 emphasizes more on cosine similarity as we focus more on trends and evolution patterns.

For retrieval, we exploit a Hierarchical Series-level Navigable Small World (HSNSW), which slightly modifies the HNSW from **Faiss** (Johnson et al., 2019) into series level with a series-level similarity measurement $\mathrm{Dist}(a, b)$. The core idea of HSNSW is to organize different series into a multi-layered graph, where each layer is a network of small world, with upper layer providing a quick search and the lower layer providing a more fine granular search (Malkov & Yashunin, 2018; Zhong, 2020). We consider the series-level affinity as the hybrid similarity in Eq. 2, and implement the series clustering to construct the layer-wise network. Then the advantage of HSNSW lies in its efficient hierarchical search from coarse to fine granularity and its scalability for dynamically inserting or deleting the records in **dataset** $\mathcal{D}$ without re-constructing the searching indexes. Given the targeted series $\boldsymbol{X}^T$, we select the Top-$K$ affinity series by Eq. 2,

$$\mathbb{X}^R = \{\boldsymbol{X}_1^R, \boldsymbol{X}_2^R, \ldots, \boldsymbol{X}_K^R\} = \underset{\mathrm{Min-K}}{\mathrm{Dist}}(\boldsymbol{X}^T, \boldsymbol{X}_i^R) \tag{3}$$

where the retrieved $K$ series are further processed as representation for constructing the prompts in ZeroTS prediction.

## 4.2 ReinLLM Adapter

Existing LLM-based solutions with RAG usually consider the retrieved series as prompts and directly take them into LLMs parallelly without any correspondence and learnable mechanism, leading to lacking of data-model interactions for model adaptation (Jin et al., 2023; Zhou et al., 2023a). In this work, with the help of time series patterns from constructed series **dataset**, ZeroTS leverages external knowledge to prompt the LLM for better prediction. We propose an RAG-LLM adapter, designated as ReinLLM, a lightweight reinforcement model, which not only introduces learnable parameters to cohere the LLM with retrieved and targeted series representations, but also allows data-model interactions by introspecting the output errors of LLM. For zero-shot prediction, the core task can be divided into two-fold. First, how to sufficiently capture the previous evolution patterns of targeted series, while second one is how to identify the underlying evolution trends and patterns of targeted unseen series by maximumly exploiting the auxiliary series. With above analysis, as illustrated in Figure 3, we model our Adapter as a reinforcement architecture equipped with a small scale learnable module, instantiated with a policy network for action selection and a value network for deriving the rewards.

**Policy network for action selection.** We determine our policy network by considering the important elements during learning as actions and update the representation with selected action as the

new states. Our policy network is composed of two branches, i.e., temporal kernel selection for target series representation and series fusion coefficient selection for retrieved series representation. Regarding target series modeling, we aim to output a series of best temporal convolution sizes as actions. We take the available targeted time series $\boldsymbol{X}^T$ as inputs, then they are fed into a temporal convolution layer with an initialized convolution kernel size $\tau$, and an MLP is followed with Softmax. The action derivation process can be formulated as, i.e.,

$$\boldsymbol{H} = \mathrm{Conv}_\tau(\boldsymbol{X}^T; \boldsymbol{W}_{cov}) \tag{4}$$

where $\boldsymbol{H}$ is the hidden representation of the target series. Assuming there are potentially $l$ convolution kernels for selection, then an ArgSort $\{\mathrm{Softmax}(\cdot)\}$ is imposed to derive the best action via descending order,

$$[\tau_1, \tau_2, ..., \tau_l] = \mathrm{ArgSort}\{\mathrm{Softmax}(\mathrm{MLP}(\boldsymbol{H}; \boldsymbol{W}_{ker})\} \tag{5}$$

The kernel associated with largest probability $\tau_1$ can be selected to implement the convolution for obtaining series-level representation. Our end-to-end learning process can help our policy network select the optimized kernel size for series representation.

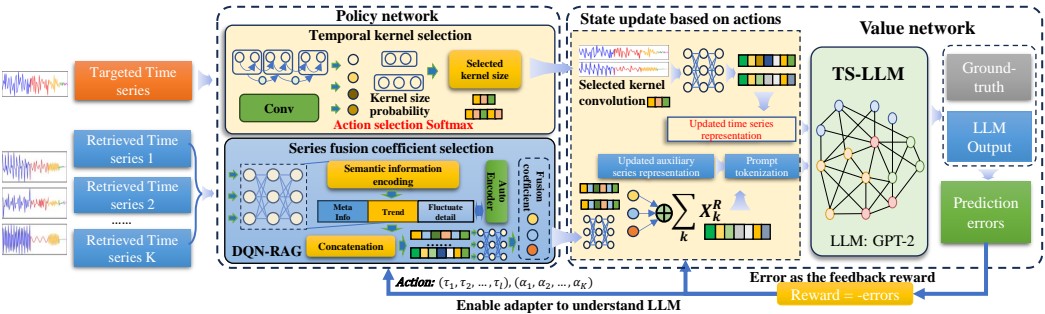

Figure 3: Illustration of ReinLLM adapter.

Regarding the policy network of series retrieval branch, we aim at extracting sufficient cues for target series and finding out the optimal fusion coefficients for $K$ retrieved series $\mathbb{X}^R = \{\boldsymbol{X}_1^R, \boldsymbol{X}_2^R, ..., \boldsymbol{X}_K^R\}$ with reinforcement learning. First, for pattern extraction of retrieved series, we construct a triple unit including three subsections, i.e., domain category, trend pattern, text description, as well as series representation, to maximumly extract sufficient cues from available series. In detail, the meta information is tokenized and concatenated by textual attributes into $\widetilde{X}_m$. The trend pattern is extracted by element-wise differential vector $(\widetilde{\boldsymbol{X}}_t)_i = \{\boldsymbol{X}_i^R(t+1) - \boldsymbol{X}_i^R(t)\}$ [1], where $t+1$ and $t$ are two adjacent temporal steps. Since detailed data description of time-series can provide informative knowledge for their regularities, such as 'traffics on rainy days' indicating more congestions and accidents within road networks. Finally, we impose a series-level representation parameters $(\widetilde{\boldsymbol{X}}_r)_i = \boldsymbol{X}_i^R * \boldsymbol{W}_a$, where $\boldsymbol{W}_a$ is the learnable parameters for representation transformation. We further concatenate the transformed $(\widetilde{\boldsymbol{X}}_r)_i$ with previous two subsections into $\widetilde{\boldsymbol{X}}_i = [(\widetilde{\boldsymbol{X}}_m)_i, (\widetilde{\boldsymbol{X}}_t)_i, (\widetilde{\boldsymbol{X}}_r)_i]$.

To fuse $K$ auxiliary series, we take advantage of the reinforcement architecture via our policy network where the aggregation weights $\{\alpha_i\}, (i \in \{1, 2, ..., K\})$ are selected actions. Specifically, we feed the concatenated representation of each series $\widetilde{\boldsymbol{X}}_i$ into our retrieval policy network with parameters $\boldsymbol{W}_{agg}$, and output the predicted action $\{\alpha_i\}$,

$$(\alpha_1, \alpha_2, ..., \alpha_K) = \mathrm{MLP}(\mathrm{Concat}[\widetilde{\boldsymbol{X}}_1, ..., \widetilde{\boldsymbol{X}}_K]; \boldsymbol{W}_{agg}) \tag{6}$$

The $\alpha_i$ will be updated in different actions in a step-by-step manner, i.e., $\alpha_i = \alpha_i - \eta$, where $\eta$ is the step size. With well-learned actions $\{\alpha_i\}$, we can update the representation of retrieved ones $X_R$,

$$\widetilde{\boldsymbol{X}}^R = \sum_{i=1}^{K} \alpha_i \widetilde{\boldsymbol{X}}_i \tag{7}$$

---

[1] We omit the superscript $R$ for the modified sequence of $\boldsymbol{X}$.

Then $\widetilde{\boldsymbol{X}}^R$ can well indicate the patterns of series close to targets, for constructing prompts. Our end-to-end learning with error-based minimization allows the automatic optimum selection of $\alpha_i$.

**LLM-feedback Value Network.** Finally, we construct the value network to derive the potential reward for each action in the ReinLLM Adapter. It has been demonstrated that introspecting the errors and differences from outside environment can make more sense than learning from the model itself (Li et al., 2024). To this end, the ground-truth of zero-shot series, can be viewed as the open-world environment, and enable our lightweight learnable module to actively interact with the LLM and outside environment (Ground-truth), thus fully activating the power of LLM. We realize it by seamlessly integrating a parameterized learnable module before our Time Series-oriented LLM (TS-LLM). Assuming $\boldsymbol{W}_T$ and $\boldsymbol{W}_R$ are two set of parameters for respectively aligning target series and retrieved ones with TS-LLM, we can respectively update their representations, {Updated targeted series: $\widetilde{\boldsymbol{X}}^T$, Retrieved augmented series for prompts: $\boldsymbol{P}$}, with selected actions in value network,

$$\widetilde{\boldsymbol{X}}^T = \mathrm{MLP}(\mathrm{Conv}_{\tau_1}(\boldsymbol{X}^T; \boldsymbol{W}_c); \boldsymbol{W}_T) \tag{8}$$

$$\boldsymbol{P} = \mathrm{MLP}(\widetilde{\boldsymbol{X}}^R; \boldsymbol{W}_R) \tag{9}$$

Then we take the errors of LLM predictor, i.e., the negative of absolute differences between LLM output and ground-truth, and such negative values can be considered as the reward of our learning process. The higher the values are, the performances are better. Specifically, we exploit the action learned from Policy Network, and update the state by re-computing the series-level representation of targets and retrieved auxiliary ones. Therefore, all learnable parameters tend to be optimized to achieve best actions and competitive prediction performances.

Regarding the large language model (LLM), we adopt GPT-2 as the backbone to avoid the label legacy and heavy training burdens, where GPT-2 is developed by OpenAI in 2019, and all testing datasets in our work is published after 2019. To enable the cooperation between series and potential prompts, we take targeted representation and prompts respectively as inputting steps $\widetilde{\boldsymbol{X}}^T$ and informative prompts $\boldsymbol{P}$. We can then obtain the following conditional output from TS-LLM,

$$\widehat{\boldsymbol{Y}}^T = \mathrm{TS} - \mathrm{LLM}(\widetilde{\boldsymbol{X}}^T | \boldsymbol{P}) \tag{10}$$

The $\widehat{\boldsymbol{Y}}^T = \{\widehat{y}_1^T, ..., \widehat{y}_Q^T\}$ is the final prediction of our ZeroTS learning system. To calibrate the prediction results, we compare the output of LLM against ground-truth of time-series, and take such differences based on MAE back to our value network and optimize the representation thus the learnable parameters can be led to approaching the LLM.

$$\mathrm{Value}(\widetilde{\boldsymbol{X}}^T, \boldsymbol{P}) = -\mathrm{MAE}(\widehat{\boldsymbol{Y}}, \boldsymbol{Y}) \tag{11}$$

The higher MAE leads to lower value and vice versa. Then our value network updates the parameters to estimate the value with learnable parameters, and impose the introspection whether the learnable small module activate the maximum power of LLM for predicting reasonable and true values approaching ground-truth. With prompts $\boldsymbol{P}$, we can easily suppress the hallucination of LLMs.

## 5 EXPERIMENT

We conduct extensive experiments on various datasets under the settings of both zero-shot and long-term prediction to verify the effectiveness of ZeroTS, where long-term prediction can be considered as a scenario with potential distribution shifts.

### 5.1 DATASETS

**Datasets for retrieval (RAG).** Since time-series for real-world usually share commonality among each other, we thus construct the **database (dataset)** from three large-scale datasets for retrieving series-level regularity, supporting our zero-shot forecasting. **a) UCR:** It is a dataset for time-series classification, including 128 subsets and covering the areas from medical science, and electricity, to geography (Tan et al., 2020). **b) Monash:** A comprehensive dataset, consisting of 30 subsets of time series, with totally more than 100,000 time series. It covers fields of health, retail, ride-sharing, and demographics (Tan et al., 2020). **c) TSB-UAD:** TSB-UAD contains 18 subsets from real-world

Table 1: Performance comparison on zero-shot forecasting

| | ETTh1→ETTh2 | | ETTh1→ETTm2 | | ETTh2→ETTh1 | | ETTh2→ETTm2 | | ETTm1→ETTh2 | | ETTm1→ETTm2 | | ETTm2→ETTh2 | | ETTm2→ETTm1 | |
|---|---|---|---|---|---|---|---|---|---|---|---|---|---|---|---|---|
| Metrics | MAE | MSE | MAE | MSE | MAE | MSE | MAE | MSE | MAE | MSE | MAE | MSE | MAE | MSE | MAE | MSE |
| TimesNet | 0.431 | 0.421 | 0.361 | 0.327 | 0.621 | 0.865 | 0.376 | 0.342 | 0.454 | 0.457 | 0.354 | 0.332 | 0.443 | 0.435 | 0.567 | 0.769 |
| Time-LLM | 0.387 | 0.353 | 0.340 | 0.273 | **0.474** | **0.479** | 0.341 | 0.272 | **0.412** | **0.381** | 0.320 | **0.268** | 0.400 | 0.354 | **0.438** | **0.414** |
| GPT4TS | 0.422 | 0.406 | 0.363 | 0.325 | 0.578 | 0.757 | 0.370 | 0.335 | 0.439 | 0.433 | 0.348 | 0.313 | 0.443 | 0.435 | 0.567 | 0.769 |
| LLMTime | 0.708 | 0.992 | 0.869 | 1.867 | 0.981 | 1.961 | 0.869 | 1.867 | 0.768 | 0.992 | 0.869 | 1.867 | 0.708 | 0.992 | 0.984 | 1.933 |
| DLinear | 0.488 | 0.493 | 0.452 | 0.415 | 0.574 | 0.703 | 0.386 | 0.328 | 0.475 | 0.464 | 0.389 | 0.335 | 0.471 | 0.455 | 0.537 | 0.649 |
| PatchTST | 0.405 | 0.380 | 0.360 | 0.314 | 0.513 | 0.565 | 0.365 | 0.325 | 0.438 | 0.439 | **0.296** | 0.334 | 0.409 | 0.425 | 0.568 | 0.492 |
| ZeroTS | **0.379** | **0.350** | **0.327** | **0.272** | 0.522 | 0.575 | **0.335** | **0.267** | 0.483 | 0.395 | 0.313 | 0.312 | **0.374** | **0.351** | 0.442 | 0.416 |

data science applications, which totally accounts for 12,686 time series with labeled anomalies spanning different domains with high variability of anomaly types, ratios, and sizes (Paparrizos et al., 2022b;a).

**Datasets as targeted time-series for forecasting.** We exploit ETTh1, ETTh2, ETTm1, ETTm2 as datasets for zero-shot predictions (Jin et al., 2023). Regarding long-term prediction, we take totally 8 datasets for evaluation, where we further employ additional four datasets those are benchmarking long-hrozion prediction models, i.e., Weather, ECL, Traffic, ILI (Wu et al., 2023). Details of datasets can be found in Appendix. A.4.

## 5.2 BASELINES

**Zero-shot forecasting.** We exploit 6 strong baselines for evaluating the zero-shot prediction setting. **1) TimesNet.** It transforms the one-dimension time-series into a two-dimension series and employ 2D-convolution for temporal pattern extraction (Wu et al., 2022). **2) Time-LLM.** A state-of-the-art series learning model that reprogrammes large language model (LLM) to adapt for time series forecasting (Jin et al., 2023). **3) GPT4TS.** It is a one-fits-all model, that exploits the frozen parameters from pretraining in computer vision and NLP to adapt time series learning (Zhou et al., 2023a). **4) LLMTime.** An LLM-based time series forecaster without inputting retrieved series as auxiliary information (Gruver et al., 2024). **5) DLinear.** A linear model for long-term series forecasting with a rolling prediction strategy. **6) PatchTST.** This baseline simultaneously explores the patch division and channel independence for multi-variate series learning (Nie et al., 2022). Among them, Time-LLM (Jin et al., 2023) achieves the best competitive performances under zero-shot predictions.

**Long-horizon forecasting.** Given that LLMTime is not utilized to perform long-term prediction, we additionally take **AutoFormer** (Wu et al., 2021) and **Informer** (Zhou et al., 2021) as baselines for evaluating long-horizon prediction. These two baselines are inherited from Transformer (Vaswani, 2017) with respectively considering auto-correlation for long-term series forecasting and probabilistic sparse self-attention.

## 5.3 IMPLEMENTATION DETAILS

**Setup for zero-shot forecasting.** In our evaluation, we follow the settings in existing literature (Jin et al., 2023). Specifically, we train our model with retrieved series on dataset $\mathcal{D}_i$ and observe how it performs on another dataset $\mathcal{D}_j$. Our ZeroTS will not see any sample in $\mathcal{D}_j$ during training period. We enable our model to output the prediction horizons over respective {96, 192, 336, 720} steps and report the average MAE and MSE for comparisons.

**Setup for long-term forecasting.** The long-term prediction set as training and testing on the same dataset with augmented series. We adopt 8 datasets for an extensive evaluation. The input length of time series is 512, and also output the horizons of respective {96, 192, 336, 720} steps for all datasets except for ILI. Since ILI is a dataset with a smaller scale, we set the forecasting horizons $Q \in \{24, 36, 48, 60\}$ for it.

## 5.4 PERFORMANCE COMPARISON

The prediction results for zero-shot and long-term settings are respectively shown in Table. 1 and 2. The best results are **bold** and the second best are underlined for comparison.

**Zero-shot prediction.** As shown, our ZeroTS achieves either best or second best performances on all 8 settings, and the improvement is ranging from 2.07% to 6.5%. Compared with well-known

Table 2: Performance comparison on long-term forecasting

|  | TimesNet | | Time-LLM | | GPT4TS | | DLinear | | PatchTST | | AutoFormer | | Informer | | ZeroTS(Ours) | |
|---|---|---|---|---|---|---|---|---|---|---|---|---|---|---|---|---|
| Metrics | MAE | MSE | MAE | MSE | MAE | MSE | MAE | MSE | MAE | MSE | MAE | MSE | MAE | MSE | MAE | MSE |
| ETTh1 | 0.450 | 0.458 | 0.423 | 0.408 | 0.455 | 0.465 | 0.437 | 0.422 | 0.430 | 0.413 | 0.487 | 0.496 | 0.795 | 1.040 | **0.421** | **0.403** |
| ETTh2 | 0.427 | 0.414 | 0.383 | 0.334 | 0.412 | 0.381 | 0.446 | 0.431 | 0.379 | 0.330 | 0.459 | 0.450 | 1.729 | 4.431 | **0.375** | **0.315** |
| ETTm1 | 0.406 | 0.400 | **0.372** | **0.329** | 0.403 | 0.388 | 0.378 | 0.357 | 0.380 | 0.351 | 0.517 | 0.588 | 0.734 | 0.961 | 0.377 | 0.347 |
| ETTm2 | 0.333 | 0.291 | **0.313** | **0.251** | 0.339 | 0.284 | 0.333 | 0.267 | 0.315 | 0.255 | 0.371 | 0.327 | 0.810 | 1.410 | 0.331 | 0.276 |
| Weather | 0.287 | 0.259 | 0.257 | 0.225 | 0.270 | 0.237 | 0.300 | 0.248 | 0.264 | 0.225 | 0.382 | 0.338 | 0.548 | 0.634 | **0.250** | **0.225** |
| ECL | 0.295 | 0.192 | **0.252** | **0.158** | 0.263 | 0.167 | 0.263 | 0.166 | 0.252 | 0.161 | 0.338 | 0.227 | 0.397 | 0.311 | 0.260 | 0.160 |
| Traffic | 0.336 | 0.620 | 0.264 | 0.388 | 0.294 | 0.414 | 0.295 | 0.433 | 0.263 | 0.390 | 0.379 | 0.628 | 0.416 | 0.764 | **0.261** | **0.384** |
| ILI | 0.931 | 2.139 | 0.801 | **1.435** | 0.903 | 1.925 | 1.041 | 2.169 | 0.797 | 1.443 | 1.161 | 3.006 | 1.544 | 5.137 | **0.795** | 1.436 |

LLM-based Time-LLM, GPT4TS, PatchTST and LLMTime, ZeroTS achieves comparative and even slightly better performances than them without any fine-tune on LLM. Along with complexity and efficiency comparison in Table. 5, we believe ZeroTS is a superior lightweight zero-shot prediction model, benefiting from retrieved augmented representation, and our actively feedback scheme.

**Long-term prediction.** In Table. 2, our ZeroTS surpasses most other solutions with Retrieved Augmented Series, achieving 5 out 8 best and 2 second best. It is worth noting that ZeroTS is with fairly fewer learnable parameters to achieve such satisfactory results. And it is also interesting to find the improvement over long-term setting is less significant than zero-shot ones. It is reasonable that ZeroTS is more effective on data scarcity scenario, where retrieved ones provide addition hints to constrain and guide LLM generation.

## 5.5 ABLATION STUDY

To investigate the superiority of each well-designed component, we devise various ablative variants via replacing the component with vanilla one or directing removing corresponding one.

**1) ZeroTS-RAG.** We remove the auxiliary sequence retrieval and exploit the LLM for zero-shot forecasting to confirm the motivation of RAG exploitation. This scheme potentially degenerates to the LLMTime (Gruver et al., 2024). **2) ZeroTS-Para.** We remove the learnable parameters in our adapter and only utilize the concatenation of retrieved series as prompts for TS-LLM. This variant also degenerate it into a model without feedback from ground-truth. **3) ZeroTS-Rein.** To verify the effectiveness of coupling reinforcement learning and LLM, we degenerate the policy network into an end-to-end learning scheme with immediate error back-propagation.

**Main results.** We perform our ablation study on two zero-shot settings in Table. 3 to verify the rationales of each design in ZeroTS. Most significantly, by removing the retrieval augmented series, our ZeroTS-RAG degenerates to an LLM-based model without any augmentation except for inputting targeted series. The performance experiences the largest drop, ranging from 9.63% to 14.76% on MAE and MSE, it suggests that retrieved series explicitly provide additional signals and regularity to help prediction, es-

Table 3: Ablation studies on zero-shot settings

|  | ETTh1 → ETTh2 | | ETTh1 → ETTm2 | |
|---|---|---|---|---|
| Metrics | MAE | MSE | MAE | MSE |
| ZeroTS-RAG | 0.435 | 0.398 | 0.364 | 0.301 |
| ZeroTS-Para. | 0.407 | 0.375 | 0.349 | 0.282 |
| ZeroTS-Rein | 0.412 | 0.380 | 0.351 | 0.286 |
| ZeroTS | **0.379** | **0.353** | **0.332** | **0.274** |

pecially on zero-shot predictions. With learnable parameters removed, our ZeroTS cannot receive the feedback from LLM and cannot be trained in an end-to-end manner, leading to lacking flexibility of correspondence between learnable module, LLM, as well as learnable module and retrieved data. Then our ablative Zero-Para has 2%-6% performance degeneration, which verifies the importance of imposing feedback from ground-truth for optimizing learnable module. In addition, Similarly, when modifying the reinforcement architecture into an ordinary end-to-end architecture, our learning module cannot find a more optimized action thus becomes inferior to integrated ZeroTS, demonstrating the rational promotion of reinforcement scheme.

## 5.6 HYPERPARAMETER STUDY AND DETAILED ANALYSIS

We provide three important hyperparameters for observing how can we obtain their best performances. 1) The number of retrieved time series from auxiliary series dataset $K = \{3, 6, 9, 12\}$. 2) Representation dimension of targeted time series ranging from $\{8, 16, 24, 32, 64\}$. 3) Representa-

tion dimension of retrieved time series (meta information and time series observations) ranging from $\{8, 16, 24, 32, 64\}$. Due to the space limitation, we only illustrate the adjustment process on zero-shot setting over ETTh1→ETTm1 on Figure. 4. As observed, $K = 6$ achieves best results, as more retrieved series will tend to involve more noise while fewer series will become less informative. For dimension of representations, we can see larger dimensions can result in better performances but we have to make the trade-off between performances and efficiency. We thus choose $K = 6$, dimensions for target and retrieved ones are set as 64.

## 5.7 CASE STUDY

To illustrate how retrieved series help better prediction, we provide intuitive analysis of intermediate results during prediction process. Given a targeted series on ETTh2, as shown in Figure. 5, our TS-RAG in ZeroTS implements a retrieval and to select Top-6 series with most proximity where three series are illustrated. The retrieved series

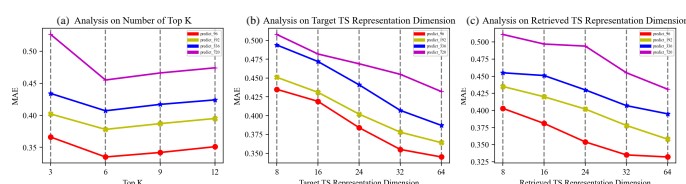

Figure 4: Hyperparameter analysis on ETTh1→ETTm1.

exactly share similar regularity and evolution patterns through the hybrid metric of proximity measurement and the similarities with targets are marked. To enhance forecasting, the auxiliary series are aggregated into a representation and we tokenize the representation as prompts into LLM, along with main input of target series. We visualize the LLM output with and without (w/o) the series for an intuitive comparison. Specifically, prediction with RAG reveals the averaged pattern of retrieved series thus overcoming the overfitting on targeted series and become smoother, while prediction w/o RAG is with more fluctuations and many details against facts. This also suggests RAG can suppress the hallucination of LLM on time series learning. Our analysis provides further interpretability and understanding of ZeroTS and highlight the contribution of RAG enhanced zero-shot prediction.

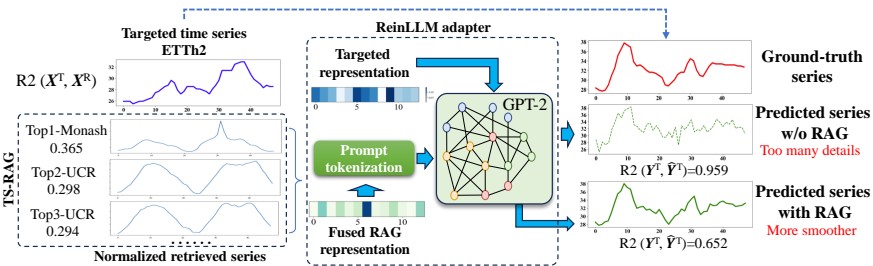

Figure 5: Case studies on ETTh2 prediction.

## 6 CONCLUSION AND FUTURE WORK

In this work, we propose a novel time-series learning framework ZeroTS, which enables multi-party model-data interactions, including retrieved series, targeted series, LLM as well as learnable adapter in an end-to-end learning manner. Specifically, we provide a series **dataset construction** protocol, a small-scale learnable adapter for cohering RAG and LLM, and the first reinforcement architecture for zero-shot time series prediction considering environment feedback. Extensive evaluation results suggest the effectiveness and superiority of our ZeroTS under zero-shot setting as well as complex long-term forecasting. We believe ZeroTS is an interesting work that provides new insights into series learning under extreme scenarios such as data scarcity, long-horizion and zero-shot predictions with small learnable parameters for modeling interactions. Future works can respectively lie in generalizing our solution to more complex scenarios as well as efficient retrieved series towards better prediction. Detailed discussion can be found in Appendix. A.5.

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

# A  Appendix

In this section, we provide more analysis regarding specific techniques, theoretical details, experimental results and efficiency comparison, to further support the superiority of our ZeroTS and help authors and reviewers better understand the operation mechanism of our idea. Finally, a brief discussion on our future work is elaborated. The codes of our ZeroTS are available at https://anonymous.4open.science/r/ZeroTS-7F7E/.

## A.1  Technical details

In this subsection, we will provide substantial technical details for better understanding the utilized techniques of ZeroTS.

### A.1.1  The architecture of ZeroTS

**In this subsection, we provide a more detailed discussion on the architecture of ZeroTS.**

Actually, since our ZeroTS aims to realize the data-model interactions, enabling our model to extract external series and how to fuse the external series with a learning objective is a natural intuition in our task. To this end, we first propose TS-RAG to realize an efficient retrieval from external series dataset. To fuse retrieved series and maximumly activate the power of LLM with RAG in time series, we devise ReinLLM adapter, it receives retrieved series as input and interconnects with LLM to derive the results, then the errors backpropagate to learnable representations on both targets and prompts, thus guiding the fusion of retrieved time series and activating the maximal power of LLM. The overview and data flow illustration are in Fig. 2 and Alg. 1, where TS-RAG and ReinLLM can jointly implement the data-model interactions from data and model sides, activating the output of LLM towards the ground-truth.

Specifically, given a targeted time-series, we activate the retrieval process with an efficient heuristic ($\rho$-$\rho$-cut-off) and hierarchical (Hierarchical Series-level Navigable Small World, HSNSW) strategy to obtain a series of auxiliary series for prompt construction.

**Regarding ReinLLM**, our **policy network** consists of two major components, i.e., temporal kernel selection for target series representation and auxiliary series fusion coefficient selection for time-series fusion. It takes the role of selecting the critical hyperparameters for targeted time-series representation and auxiliary time series fusion. **Update process.** When the action (important hyperparameter) selected, the representation will be updated with new hyperparameters, and then feed the updated representation into LLM, where targeted representation as main input to LLM and fused auxiliary series as prompt to LLM for supplementary cues. After that, TS-LLM, instantiated by GPT-2, outputs the prediction results and deliver the outcome and errors to the value network. Our **value network** transfers the errors into reward and backpropagate to the policy network for strategy optimization. Actually, our lightweight learnable module is optimized from both data side and model side. On data side, we exploit retrieval process from auxiliary real-world data, and let updatable real data to continuously supplement the cues for model prediction. And on model side, the model (LLM) is considered as the environment and we let our lightweight learnable module ReinLLM interact with LLM and derive the error feedback as reward to optimize the whole process, contributing to the real-world corrections and enabling seamless data-model-world interactions.

### A.1.2  Structure of RAG dataset and how it utilize for forecasting

Here, we provide a table to illustrate the detailed description of this key-value based structural retrieval dataset for large-scale time series, as shown in Table. 4.

**Utilization of meta information for retrieval.** The key-value in Table.4 can be considered as the meta information. In this subsection, we provide details on how we utilize Regarding the retrieval process, we utilize the meta information including domain category (e.g., traffics/electricity/weather. . . ), timestamps (2024/11/11/10:00) and spatial location/user identification to help retrieve related auxiliary series. Of course, calculated mean and variance are also considered meta information for similarity-based retrieval. Specifically, given a targeted series with its domain category, timestamps, and identification (user/location id.) where it is available when data is collected, we can retrieve time series from auxiliary series base by indexing corresponding meta

information (keywords), if the meta information is matched, e.g., two series share the same domain category, share similar timestamps (day of week, hour of day), this series will be selected. And we can compute the similarity between selected series and targeted series, where the top-K similar series will be utilized for following fusion and representation as prompts.

**Meta information utilization in ReinLLM.** In ReinLLM, meta information is considered as the features of retrieved series, which consists of domain category, timestamps and data description, and we take the concatenation of meta information, trend vector as well as deterministic series values as the integrated representations. The abovementioned meta information (domain category, timestamps and data description) is available when the auxiliary base is constructed, and it corresponds to the dataset and series-level property. Then the words are transferred into vectors with the tokenizer in LLM (i.e., GPT-2 in our ZeroTS), where the tokenizer is trained by the strategy named Byte-Pair Encoding (BPE), i.e., the tokenizer trained with occurrence frequency of words and their semantics.

### A.1.3 COMPLEXITY ANALYSIS OF RETRIEVAL

Assume there are $M$ times of computation for retrieving the matched Top-$K$ series. To reduce the complexity to a reasonable range, we propose a $\rho - \rho$-cut-off over sampling and dimensions, thus the full computational loads are reduced to $\rho^2 M$. Regarding retrieved series, we every time sample $\rho\%$ of overall all samples for similarity comparison with a stratified sampling strategy, thus the pair-wise similarity computations are reduced. The stratified strategy ensures such sampling with fairness and diversity. For representation dimension, we further reduce the dimension of representation to $\rho\%$ of original vectors, then the computing loads are further significantly decreased.

### A.1.4 CLARIFICATION ON POTENTIAL INFORMATION LEAKAGE

We have taken the issue of information leakage into consideration with the following two designs.

- First, to avoid the information leakage of LLM, and fairly investigate the effectiveness of our idea, we exploit GPT-2 (released in 2019) instead utilizing a most recent model as backbone, and the testing/inference datasets are all released after 2019. That's to say, the LLM has not seen the training/testing set when experiencing the pre-training.

- Second, to avoid the information leakage of RAG, we utilize the disjoint datasets in retrieval ones and inference ones and ensure the ordered timestamps in the retrieval. More specifically, in our testing stage, to ensure fair testing and imitate the real-status, we filter the available series with meta information ('timestamps') and directly mask the 'future observations' corresponding to this targeted series as unseen ones (the remaining $1 - \rho$). To this end, we can ensure our solution undergo a fair and trustworthy test. And in deployed real-world applications, if we are expected to predict future steps, actually only previous events and observations are available as no one can reach the future beyond the present.

With above well-designed strategies, we can declare that our solution will not encounter the information leakage in both real applications and testing periods.

### A.1.5 DISCUSSION ON SELECTION OF AUXILIARY DATASET

As discussed in our main text, our model ZeroTS does not require intensive domain-related data but exploit the intuition in our Fig.1 that series share similar trends in the world can reinforce each other. To this end, we further provide some discussions on how can we address challenges when similar series with targets are limited. As we know, from a classical signal decomposition theory Akansu & Liu (1991); Akansu & Haddad (2001), time-series tends to be decomposed of different components, or a complex series can be the combinations of several simple components. Thus, we can take this theory to interpret our auxiliary series based solution. Our retrieval and fusion take a Top-K selection mechanism and then dynamically learn the fusion coefficients with reinforcement scheme. Even though series have limited direct similarity, their combinations may make sense. In our solution, we can always find the Top-K series and dynamically combine them in a learnable manner and further provide the cues of potential trends for output of the backbone language model. Regarding data selection, there must be unavoidable performance variations on different collection of dataset. Our intuition is that when the scale and diversity of external series set is increasing, the more patterns are included, but we cannot numerate all time-series in the world and finding additional series will also

Table 4: Structural key-value instance

| | Meta information | | | | | Deterministic observation |
|---|---|---|---|---|---|---|
| Key | Domain | Timestamps | Spatial/User ID | Mean | Variance | Series value |
| Value | Electricity | 20140101 | (119.52E, 31.05N) | 0.35 | 0.42 | [0.32,0.34,0.45, . . . ] |

lead to further unexpected complexity and burden on collection. The perspective of our solution is to alleviate the influences of variations on external dataset selection and mine the potential relationships between retrieval series and target ones with trade-off. Moreover, for reducing complexity and selecting more related series, clustering external series and using prototype series for pre-selection can be investigated as future works.

---

**Algorithm 1** The training process of ZeroTS

---

**Input:** Target series $\boldsymbol{X}^T$, world-wide series external dataset $\mathcal{D}$.
**Output:** Retrieved Top-K series $\mathbb{X}^R = \{\boldsymbol{X}_0^R, \boldsymbol{X}_1^R, ..., \boldsymbol{X}_K^R\}$, Predicted target series $\widehat{\boldsymbol{Y}}^T = \mathcal{M}^*(\boldsymbol{X}^T | \mathbb{X}^R)$.
**Initial:** the TS-LLM model, learnable ReinLLM adapter, the number of total epochs M.
**for** $i = 1$ **to** $M$ **do**
    Retrieve time-series from databse $\mathbb{X}^R = \text{TS} - \text{RAG}(\boldsymbol{x}_i), \mathbb{X}^R \in \mathcal{D}$
    Run Policy Network in ReinLLM
    $[\tau_1, ..., \tau_l] = \text{ArgSort}\{\text{Softmax}(\text{MLP}(\boldsymbol{H}; \boldsymbol{W}_{ker})\}$
    $(\alpha_1, ..., \alpha_K) = \text{MLP}(\text{Concat}[\widetilde{\boldsymbol{X}}_1, ..., \widetilde{\boldsymbol{X}}_K]; \boldsymbol{W}_{agg})$
    Run Value Network in ReinLLM
    $\widetilde{\boldsymbol{X}}^T = \text{MLP}(\text{Conv}_{\tau_1}(\boldsymbol{X}^T; \boldsymbol{W}_c); \boldsymbol{W}_T); \boldsymbol{P} = \text{MLP}(\widetilde{\boldsymbol{X}}^R; \boldsymbol{W}_R)$
    Feed into TS-LLM, $\widehat{\boldsymbol{Y}}^T = \text{TS} - \text{LLM}(\widetilde{\boldsymbol{X}}^T | \boldsymbol{P})$
    Compute the reward $\text{Value}(\widetilde{\boldsymbol{X}}^T, \boldsymbol{P})$
    **Optimization**
**end for**
**Return** $\mathbb{X}^R$, Predicted $\boldsymbol{Y}^T$

---

### A.1.6 ALGORITHM OF HSNSW

Efficient series retrieval is vital for RAG as well as our zero-shot learning. Here, we exploit the Hierarchical Series-level Navigable Small World (HSNSW) algorithm to help implement our retrieval, where we slightly modify HNSW into a series-level retrieval with a hybrid similarity metric of Eq. 2. **Here we briefly introduce the architecture of HNSW. For HNSW, its core idea is to organize different samples into a multi-layered graph, where each layer is a network of small world, with upper layer providing a quick search and the lower layer providing a finer granular search (Malkov & Yashunin, 2018; Zhong, 2020). We then let it modify into a series-level retrieval process, name this mechanism as Hierarchical Series-level Navigable Small World (HSNSW). Actually, we implement it by the Faiss which is an open-sourced package developed by Facebook (Johnson et al., 2019).**

Specifically, it involves several key parameters and probability calculations in its mathematical formulation and construction process. This section outlines some of the core mathematical concepts and formulas that underpin the HNSW algorithm. We illustrate it in Figure. 8.

**Layer Probability Calculation.** During the construction process, the insertion probability of each vector into different layers is given by the following probability function,

$$\text{Prob}(\text{level } l) = \exp\left(-\frac{l}{m_L}\right) \times \left(1 - \exp\left(-\frac{1}{m_L}\right)\right) \tag{12}$$

where $m_L$ is the level multiplier used to balance the distribution of vectors across different layers. $m_L = 0$ indicates that all vectors are inserted only at layer 0.

**Optimal Level Multiplier.** To minimize the overlap of shared neighbors across layers and balance the average number of traversals during the search process, the rule of thumb for the optimal level

multiplier is:

$$m_L = \frac{1}{\ln(M)},$$

where $M$ is the number of neighbors each vertex has.

**Neighbor Selection.** At each layer, the algorithm greedily traverses edges to find the $ef$ nearest neighbors of the inserted vector $q$, with $ef$ initially set to 1. In the second phase of construction, $ef$ is increased to $ef_{\text{Construction}}$, thus returning more nearest neighbors as candidate points for links.

**Link Addition.** When adding links, two parameters $M_{\max}$ and $M_{\max 0}$ are considered, which define the maximum number of links a vertex can have at non-zero layers and at layer 0, respectively.

**Stopping Condition.** The stopping condition for the insertion operation is to find a local minimum at layer 0, i.e., no closer neighbors are found at this layer.

## A.2 DISTINGUISH ZEROTS AGAINST OTHER RELATED WORKS

As ZeroTS is at the intersection of reinforcement learning, utilization of LLM and time-series learning, we then provide a systematical discussion regarding RLHF, LLM choice and why not fine-tune the LLM to enhance and verify our design motivations.

### A.2.1 DISCUSSIONS BETWEEN ZEROTS AND RLHF

Actually, both ZeroTS and RLHF (Reinforcement Learning from Human Feedback) utilize reinforcement scheme to enhance the model performance. However, they are different on which side of information to receive and how to implement further optimization. Specifically, ZeroTS receives the feedback from LLM and optimizes the representation of fused retrieved time series, where the feedback can be automatically computed via the differences between ground-truth and LLM output, and it enjoys the nice property of delivering the model side feedback to previous data side. In contrast, RLHF in InstructGPT explores a fine-tune process (PPO) to align the awareness between GPT and human annotated labels (Ouyang et al., 2022). DPO exploits a scheme to remove the unstable reinforcement process and direct optimizes the reward from human label rewards (Rafailov et al., 2024). To this end, RLHF requires intensive data and extensive human labors for LLM fine-tune, while our ZeroTS optimizes the retrieved series (both representation and fusion mechanisms) fusion with gaining awareness from LLM output side, which is more efficient, automatic, and lightweight with fewer human workloads, yielding more practicality.

### A.2.2 DISCUSSIONS ON MOTIVATION OF LLM AND THE VERSION CHOICE OF LLM

Actually, the architectures (backbones) for time-series forecasting in are all different variants of transformer, i.e., Timer (Liu et al., 2024), UniTST (Authors), MOMENT (Goswami et al.), Chronons (Ansari et al.) are transformer-based and Lag-Llama (Rasul et al.) is a recent LlaMA-based model. Noted that these above-mentioned transformer architectures require training from scratch and the parameter scales of these models range from Million level to Billion level according the scale of datasets. In contrast, our ZeroTS is a rather lightweight model compared to them, which inherits the architecture of GPT-2 without training or fine-tuning such GPT. Our model is with 50M Trainable Param with 700M of GPT-2 for inference, which is lighter than above methods. Furthermore, in fact, there is not an explicit boundary between the conventional foundation models and LLM, the full name of GPT-2 is generative pretraining Transformer at Version 2, where the inner structure is also transformer. We further conduct empirical tests to demonstrate the performance difference among LLM, conventional foundation model and a more recent LLM LlaMA-3 in Section A.4.4, where the results suggest the superior effectiveness of LLM.

### A.2.3 COMPARISON WITH FINE-TUNING MODELS

It is acknowledged that LLM without fine-tuning leads to less flexibility of language model. But actually, fine-tuning the LLM requires substantial sequence observations and large-scale computational costs, this cannot well adapt to the zero-shot time series learning scenarios. To this end, we consider detour the fine-tuning of LLM and propose a lightweight learnable module to remedy the non-fine-tuning of LLM, which allows interactions among real-world external data, LLM model,

as well as model error feedback, thus guiding the fusion of retrieved time series and activating the maximal power of LLM. We elaborate the motivation and reasons for our designs.

- **Data availability and computational costs are challenges cannot be neglected.** LLM is a well pre-trained model with large number of parameters equipped with sufficient semantic patterns within sequences and documents. Therefore, fine-tuning and impacting them will require feeding into large-scale new data and substantial computational costs. However, in in our zero-shot research, the data availability and computational power are big challenges and cannot satisfy enabling the changes of LLM. That's to say, limited available data can nearly fail to enable changes of parameters in LLM. Let alone for zero-shot learning, it is practically impossible to fine tune LLM (parameters ranging from 700M to 8B) with new arriving data in a real-time manner. Thus, we devise a more lightweight manner which directly utilize the LLM model and learn a small model for prompt construction. Regarding efficiency, we have provided analysis in Table.5 of manuscript and we can observe that our ZeroTS only consumed approximately 1/4 memory overloads and 1/7 inference speed against peer models, indicating its superior efficiency.

- **It is necessary to incorporate additional inputs as prompts to supplement sequential pattern cues.** To remedy the model adaptation capacity, we especially introduce and preserve an external time-series base (data set), which can be dynamically updated to continuously supplement the new knowledge from real world. To implement this, we couple the emerging Retrieval Augmented Generation (RAG) technique with time-series learning and devise TS-RAG with diverse efficient retrieval strategies. With dynamically updated and retrieved series that share similarity with targeted ones, the series can provide supplementary patterns and leading evolution patterns incorporated into retrieved ones for prompting LLM outputs.

- **It is necessary to devise active feedback for optimizing targeted and retrieved series representation.** To adapt model with real-world data and ground-truth and equip the model with adaptation capacity, we design a ReinLLM to cohere the feedback from LLM with prepositive series representation learning.

Overall, our ZeroTS is designed tailored for zero-shot time series learning from a new perspective of data-model interactions with a great trade-off, i.e., designing a lightweight learnable ReinLLM adapter. We believe such trade-off has well leveraged the power of LLM and allowed interactions between data and model sides, thus providing sufficient and flexible complementary to existing LLM for series forecasting.

### A.2.4 DISTINGUISH ZEROTS WITH OTHER CROSS-DOMAIN LEARNING AND MULTI-TASK LEARNING WORK

It is interesting to compare our ZeroTS with other cross-domain learning and multi-task learning work, which is also prevailing in time-series learning and spatiotemporal forecasting. It is a good idea to train cross-domain model using data from different domains, and it has been verified for its effectiveness in pioneering research of UniST (Yuan et al., 2024) and CMuST (Yi et al., 2024). We can name it as unified learning or continuous multi-task learning. For UniST (Yuan et al., 2024), it falls into the idea of training a cross-domain model using multi-sourced/domain data and it exactly demonstrates the improvement and zero-shot capacity. For training and fine-tune scheme, CMuST devises a task-level continuous learning to iteratively fine-tune the new task with partial neuron stable (Yi et al., 2024), which also provides additional capacity in zero-shot and cold-start scenarios for dynamic learning. Even so, we argue that the unified model cannot adapt all scenarios and lack flexibility to continuous update (Yuan et al., 2024) while continuous learning lacks the flexibility to retrieve any external knowledge and requires additional fine-tune parameters, limiting the zero-shot learning capacity. We think the key point of this research line is to decouple the invariant and variable parts, and transfer the invariances to new domain (task) while assimilate changes from environments.

**A toy example test on cross-domain learning.** A toy experiment on NYC (with crowd in/out and taxi pick/drop four domain series) can be obtained from Ref.ppyiget, when training on other 3 tasks, and the 4th task with only 25% samples, the cross-domain learning can improve performance of MAPE by 5% from best baseline 0.473 (PromptST) to multi-task learning setting 0.450 (cross-

Table 5: Empirical complexity comparison on long-term prediction of ETTh1

| Length | ETTh1-96 | | | ETTh1-336 | | |
|---|---|---|---|---|---|---|
| Metric | Trainable Param.(M) | Mem.(MiB) | Speed(s/iter) | Trainable Param.() | Mem.(MiB) | Speed(s/iter) |
| Llama (32)-QLoRA | 50.29 | 45,226 | 0.697 | 50.37 | 49,374 | 0.732 |
| Llama (32)-Reprogram | 6.39 | 32,136 | 0.517 | 6.48 | 37,988 | 0.632 |
| ZeroTS (GPT2) | 53.44 | 8,244 | 0.064 | 55.51 | 8,416 | 0.096 |

domain). Even so, cross-domain learning still suffers two critical issues 1) only correlated tasks and domains can be utilized to reinforce each other and blindly involving various tasks can lead to model suffering noise and distractions for main predictions, 2) the unified model may not accommodate patterns from all tasks that limits the extendibility.

**Summary of discussions against other related works.** To conclude, our ZeroTS is a totally different perspective to address zero-shot and generalization task, which detours the large model fine-tune, and is with more flexibility and efficiency without considering task-level or domain-level similarity. Our solution allows any time-series to be retrieved, and incorporated into the model dynamic update, utilizes the additional temporal evolution cues (mostly similar to target one) to prompt the LLM for output prediction. ZeroTS can simultaneously exploit the power of external knowledge in a plug-and-play manner and informative semantics and regularity in LLM in a lightweight manner.

### A.3 THEORETICAL ANALYSIS AND COMPLEXITY EFFICIENCY ANALYSIS

In this subsection, we provide some theoretical analysis from information theory and complexity efficiency analysis to demonstrate the effectiveness and efficiency in a formal manner.

#### A.3.1 EFFECTIVENESS OF RAG.

We provide a brief but powerful analysis from the perspective of information theory. Assume all learnable parameters $\Theta = \{\boldsymbol{W}_i\}$, the targeted series $\boldsymbol{X}^T = \{\boldsymbol{X}_1^T, ..., \boldsymbol{X}_T^T, \boldsymbol{Y}_{T+1}^T, ..., \boldsymbol{Y}_{T+Q}^T\}$ where $\boldsymbol{X}^T$ denotes the input features while $\boldsymbol{Y}^T$ is the output target. Since entropy can measure the degree of chaos in data, which is equivalent to the learning difficulty of corresponding dataset and thus their predictability. The smaller entropy reflects less discrepancy and larger predictability (DelSole & Tippett, 2007). Then the conventional prediction model $\mathcal{M}$ maximizes the mutual information between the input previous series of $H(\boldsymbol{X}^T, \boldsymbol{Y}^T)$.

When RAG is introduced for zero-shot prediction, the retrieved series $\mathbb{X}^R = \{\boldsymbol{X}_1^R, ..., \boldsymbol{X}_K^R\}$ can be considered as the informative knowledge from an external database (dataset). If the similarity between targeted series and retrieved ones $\mathrm{Dist}(\boldsymbol{X}_i^R, \boldsymbol{X}^T) > th_{sim}$ where $th_{sim}$ is a significant threshold indicating the overlapping patterns between retrieved and target series, we can derive the objective of $\mathcal{M}^*$ is equivalent to minimize the entropy $H(\boldsymbol{X}^T, \boldsymbol{Y}^T | \mathbb{X}^R)$. According to the condition reduction principle in entropy (Ash, 2012), we first have $H(\boldsymbol{X}) > H(\boldsymbol{X} | \mathbb{X}^R)$ and $H(\boldsymbol{Y}) > H(\boldsymbol{Y} | \mathbb{X}^R)$. We then demonstrate the joint entropy also satisfies the condition reduction principle.

$$
\begin{aligned}
H(X, \boldsymbol{Y} | \mathbb{X}^R) &= H(\boldsymbol{Y} | \boldsymbol{X}, \mathbb{X}^R) + H(\boldsymbol{Y} | X^R) \\
&< H(\boldsymbol{Y} | \boldsymbol{X}) + H(\boldsymbol{X} | \mathbb{X}^R) \\
&< H(\boldsymbol{Y} | \boldsymbol{X}) + H(\boldsymbol{X}) \\
&= H(\boldsymbol{X}, \boldsymbol{Y})
\end{aligned}
\tag{13}
$$

We can then systematically derive that introducing the retrieved series with explicit overlapping patterns can improve the informativeness of prediction model and squash the information from data aspect into learnable parameters $\Theta$, thus enabling quickly adapting to LLM and new unseen data.

#### A.3.2 COMPLEXITY ANALYSIS.

The complexity analysis is two-fold with two stages in our ZeroTS, i.e., TS-RAG for series retrieval and training process of ReinLLM adapter.

**Theoretical analysis.** First, as discussed in above technical details, there are $M$ times of computation for retrieving the matched Top-$K$ series and the proposed $\rho$-$\rho$-cut-off over sampling and

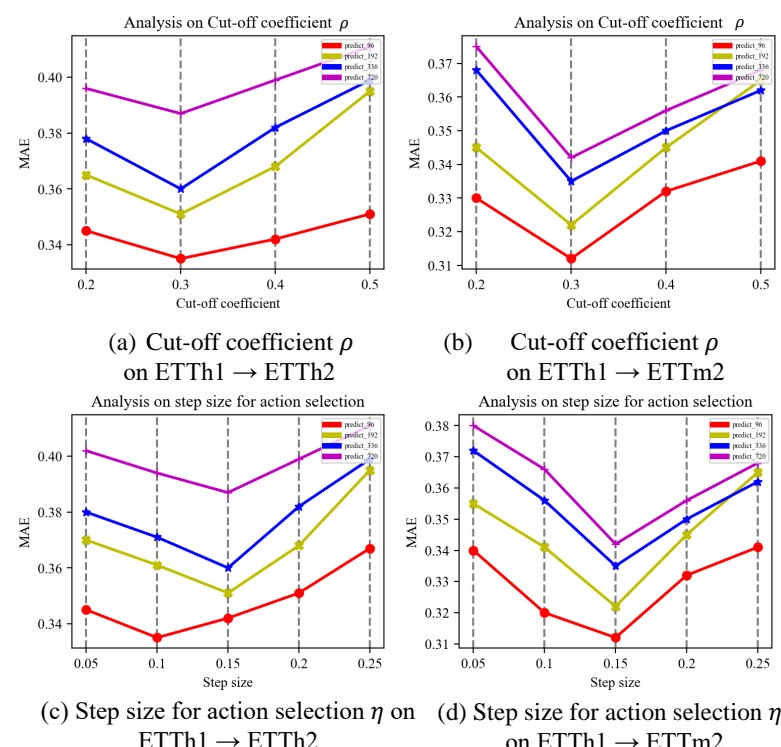

Figure 6: Analysis of cut-off coefficient $\rho$ and step size $\eta$

dimensions, thus the full computational loads have been reduced to $\rho^2 M$, where $\rho$ is expected to satisfy $0 < \rho < 0.5$. Second, considering the training process of learnable parameters in our Adapter, the trainable parameters are in a small scale, consisting of temporal convolutions and corresponding action selection, retrieved series representation, aggregation learner of retrieved series where it is noted that none fine-tune for LLM.

**Empirical analysis.** The comprehensive empirical comparisons on complexity are reported in Table. 5. We report the total number of trainable parameters (in million), GPU memory (in mebibyte, MiB), and running timeseconds per iteration (s/iter) by taking ETTh1-96 and ETTh1-336 as instances. The compared models are retrieved from TimeLLM (Jin et al., 2023). The number of parameters are slightly more than others, where the reason lies in introducing learnable scheme into retrieval of RAG, which makes the trade-off between the high-dimension retrieval process and low-dimension but learnable retrieval. Even though, **we can observe that our ZeroTS has an overwhelmingly superiority on memory and speed metrics, nearly 1/4 memory overloads and 1/7 inference speed against peer models, indicating our work has significantly improved the memory space and inference speed.** Such superiority can enable ZeroTS extremely friendly to edge computing and resource-limited services. We also believe the number of parameters can be reduced by adjusting the hyperparameters of retrieval process, and learnable weights for efficiency tradeoff.

Then we can conclude our our retrieval is efficient. Even though, we believe that efficiency is a permanent issue in retrieval and deep learning training process. In the future, we are going to investigate more efficient strategy to retrieve more accurate series as prediction prompts and reduce the learnable module for quick data adaptation.

A.4 ADDITIONAL EXPERIMENTAL RESULTS

In this subsection, we first provide the detailed datasets and metrics for evaluations, and then provide more analysis regarding hyperparameter settings.

Table 6: Horizon-wise results on zero-shot prediction

|  | MAE | TimesNet | Time-LLM | GPT4TS | LLMTime | DLinear | PatchTST | ZeroTS(Ours) |
|---|---|---|---|---|---|---|---|---|
| ETTh1→ETTh2 | 96 | 0.387 | 0.337 | 0.374 | 0.576 | 0.4 | 0.35 | 0.336 |
|  | 192 | 0.429 | 0.374 | 0.417 | 0.586 | 0.460 | 0.400 | 0.368 |
|  | 336 | 0.451 | 0.415 | 0.444 | 0.637 | 0.505 | 0.428 | 0.396 |
|  | 720 | 0.458 | 0.420 | 0.452 | 1.034 | 0.589 | 0.443 | 0.417 |
|  | Avg | 0.431 | 0.387 | 0.422 | 0.708 | 0.488 | 0.405 | 0.379 |
| ETTh1→ETTm2 | 96 0 | 0.313 | 0.293 | 0.315 | 0.563 | 0.357 | 0.304 | 0.271 |
|  | 192 | 0.342 | 0.312 | 0.342 | 0.654 | 0.413 | 0.339 | 0.31 |
|  | 336 | 0.371 | 0.365 | 0.374 | 0.728 | 0.465 | 0.373 | 0.341 |
|  | 720 | 0.419 | 0.39 | 0.422 | 1.531 | 0.573 | 0.424 | 0.387 |
|  | Avg | 0.361 | 0.34 | 0.363 | 0.869 | 0.452 | 0.36 | 0.327 |
| ETTh2→ETTh1 | 96 | 0.601 | 0.452 | 0.577 | 0.777 | 0.555 | 0.465 | 0.485 |
|  | 192 | 0.61 | 0.461 | 0.559 | 0.82 | 0.568 | 0.509 | 0.501 |
|  | 336 | 0.626 | 0.482 | 0.578 | 0.864 | 0.577 | 0.515 | 0.544 |
|  | 720 | 0.648 | 0.502 | 0.597 | 1.461 | 0.596 | 0.561 | 0.558 |
|  | Avg | 0.621 | 0.474 | 0.578 | 0.981 | 0.574 | 0.513 | 0.522 |
| ETTh2→ETTm2 | 96 | 0.324 | 0.276 | 0.329 | 0.563 | 0.336 | 0.309 | 0.261 |
|  | 192 | 0.352 | 0.315 | 0.346 | 0.654 | 0.369 | 0.345 | 0.324 |
|  | 336 | 0.383 | 0.337 | 0.376 | 0.728 | 0.397 | 0.379 | 0.365 |
|  | 720 | 0.446 | 0.417 | 0.429 | 1.531 | 0.442 | 0.421 | 0.391 |
|  | Avg | 0.376 | 0.341 | 0.37 | 0.869 | 0.386 | 0.365 | 0.335 |

### A.4.1 DATASET.

The datasets we exploit for zero-shot predictions are ETTh1, ETTh2, ETTm1, ETTm2, following literature (Jin et al., 2023). These datasets provide electricity consumption indexes of urban users, where ETTh1 and ETTh2 are collected on hour-level, while ETTm1 and ETTm2 are collected on 15-minute levels. They are adopted to be trained on one set and tested on another one, and this is a cross-transfer process within any other two datasets to imitate (implement) zero-shot settings. Regarding long-term prediction, we take totally 8 datasets for evaluation, where we further employ additional four datasets those are benchmarking long-hrozion prediction models, i.e., Weather, ECL, Traffic, ILI (Wu et al., 2023). Specifically, the dataset Weather is comprised of one-year records from 21 meterological stations in Germany with sampling rate of 10 minutes. The Traffic dataset includes 862 traffic sensors across the State of California, with a sampling rate of 1 hour. The influenza-like illness (ILI) contains records of patients experiencing severe influenza with complications.

### A.4.2 EVALUATION METRIC.

We exploit mean absolute error (MAE) and mean square error (MSE) as metrics for both zero-shot and long-horizon forecasting.

$$\text{MAE} = \frac{1}{Q}\sum_{t=T+1}^{T+Q}|Y_t - \widehat{Y}_t|; \quad \text{MSE} = \frac{1}{Q}\sum_{t=T+1}^{T+Q}(Y_t - \widehat{Y}_t)^2 \tag{14}$$

### A.4.3 PERFORMANCE COMPARISON ON HORIZONS.

Drawing the results for each prediction horizon brings more informativeness and helps better understanding of our model. Here, we report the results for each horizon on long-term forecasting and zero-shot forecasting, where MAE results of four setting are reported in respective tasks, i.e., ETTh1→ETTh2, ETTh1→ETTm2, ETTh2→ETTh1, ETTh2→ETTh1 for zero-shot forecasting, and ETTh1, ETTh2, ETTm1, ETTm2 for long-term prediction. The results can be found in Table. 6 and Table. 7.

As observed in above tables, we can conclude our solution outperforms baselines on most horizons, and there is no skewing on one horizon, which enhance the effectiveness of our ZeroTS.

### A.4.4 ABLATION ON RECENT LLAMA-3 AND PATCHTST

To further verify the motivation of exploiting LLM as a tool for prediction and the choice of GPT-2, we conduct a series of further experiments by combining RAG with respective recent Llama-3 (with 8 Billion para.), and PatchTST. We also demonstrate the performances of pure PatchTST as comparison to enhance the rationality of utilizing GPT models. The results are shown in Table. 8.

Table 7: **Horizon-wise results on long-term prediction**

| | MAE | TimesNet | Time-LLM | GPT4TS | DLinear | PatchTST | AutoFormer | Informer | ZeroTS(Ours) |
|---|---|---|---|---|---|---|---|---|---|
| ETTh1 | 96 | 0.402 | 0.392 | 0.397 | 0.399 | 0.399 | 0.459 | 0.713 | 0.381 |
| | 192 | 0.429 | 0.418 | 0.418 | 0.416 | 0.421 | 0.482 | 0.792 | 0.399 |
| | 336 | 0.469 | 0.427 | 0.433 | 0.443 | 0.436 | 0.496 | 0.809 | 0.438 |
| | 720 | 0.500 | 0.457 | 0.456 | 0.49 | 0.466 | 0.512 | 0.865 | 0.466 |
| | Avg | 0.450 | 0.423 | 0.455 | 0.437 | 0.430 | 0.487 | 0.795 | 0.421 |
| ETTh2 | 96 | 0.374 | 0.328 | 0.342 | 0.353 | 0.336 | 0.388 | 1.525 | 0.325 |
| | 192 | 0.414 | 0.375 | 0.389 | 0.418 | 0.379 | 0.452 | 1.931 | 0.373 |
| | 336 | 0.452 | 0.409 | 0.407 | 0.465 | 0.380 | 0.486 | 1.835 | 0.391 |
| | 720 | 0.468 | 0.42 | 0.441 | 0.551 | 0.422 | 0.511 | 1.625 | 0.410 |
| | Avg | 0.427 | 0.383 | 0.412 | 0.446 | 0.379 | 0.459 | 1.729 | 0.375 |
| ETTm1 | 96 | 0.375 | 0.334 | 0.346 | 0.343 | 0.342 | 0.475 | 0.571 | 0.345 |
| | 192 | 0.387 | 0.358 | 0.372 | 0.365 | 0.369 | 0.496 | 0.669 | 0.359 |
| | 336 | 0.411 | 0.384 | 0.394 | 0.386 | 0.392 | 0.537 | 0.871 | 0.384 |
| | 720 | 0.45 | 0.411 | 0.421 | 0.421 | 0.42 | 0.561 | 0.823 | 0.421 |
| | Avg | 0.406 | 0.372 | 0.403 | 0.378 | 0.380 | 0.517 | 0.734 | 0.377 |
| ETTm2 | 96 | 0.267 | 0.253 | 0.262 | 0.269 | 0.255 | 0.339 | 0.453 | 0.257 |
| | 192 | 0.309 | 0.293 | 0.301 | 0.303 | 0.292 | 0.34 | 0.563 | 0.32 |
| | 336 | 0.351 | 0.329 | 0.341 | 0.342 | 0.329 | 0.372 | 0.887 | 0.361 |
| | 720 | 0.403 | 0.379 | 0.401 | 0.421 | 0.385 | 0.432 | 1.338 | 0.387 |
| | Avg | 0.333 | 0.313 | 0.339 | 0.333 | 0.315 | 0.371 | 0.810 | 0.331 |

Table 8: **Ablation on Recent Llama-3 and PatchTST**

| MSE | RAG+Llama-3 | RAG+PatchTST | PatchTST | RAG+GPT2(ZeroTS) |
|---|---|---|---|---|
| ETTh1→ETTh2 | 0.345 | 0.361 | 0.380 | 0.350 |
| ETTh1→ETTm2 | 0.274 | 0.286 | 0.314 | 0.272 |
| ETTh1 | 0.402 | 0.410 | 0.413 | 0.403 |
| ETTh2 | 0.312 | 0.319 | 0.330 | 0.315 |

It is noted that a more powerful LLM can improve the performances maximumly by 1.42% (obtained on ETTh1→ETTh2 scenario from 0.350 to 0.345, others are ranging from 0.2% to 0.9%), but the inference efficiency of larger LM (e.g., Llama-3: 8 Billion) has dropped significantly. When replacing LLM with PatchTST, the performance will drop by 1% to 5% (maximumly arrive at ETTh1→ETTm2 from 0.272 to 0.286). We can witness the improvement of LLM is relatively deserve for its efficiency and enhance the choice of GPT-2 in ZeroTS. Thus, we can conclude that our GPT-2 is sufficient for our task and more powerful LLM can further sacrifice efficiency which may not deserving for our predictions.

### A.4.5 OUT-OF-THE-BOX ZERO-SHOT TESTS

To further verify the effectiveness, we have expanded experiments to a more general setting, i.e., with no particular task-specific exemplars for tuning. In our ZeroTS, we utilize the non-training GPT-2 with RAG (retrieval strategy TS-RAG) for implementation. ZeroTS has degenerated into a pure RAG model without learnable parameters and strategies. We also take Llama-3, PatchTST as comparisons with fair settings. Three typical datasets, i.e., ETTh1, ETTm1, and Traffic are selected to illustrate the out-of-the-box prediction results of our non-trainable version of ZeroTS. The results are illustrated in Table. 9.

Actually, to this end, our design of RAG+GPT-2 can outperform conventional model PatchTST, and achieve comparable performances with RAG+Llama-3 but with sacrificing large-scale computational loads. Therefore, we can conclude that the choice of GPT-2 is fine and superior to others regarding efficiency and effectiveness trade-off.

Table 9: **Out-of-the-box zero-shot tests**

| | RAG+Llama-3 | RAG+PatchTST | RAG+GPT2 (ZeroTS) |
|---|---|---|---|
| Full zero-shot on ETTh1 | 0.442 | 0.480 | 0.457 |
| Full zero-shot on ETTm1 | 0.440 | 0.465 | 0.443 |
| Full zero-shot on Traffic | 0.456 | 0.504 | 0.472 |

Table 10: **Significance test on zero-shot forecasting**

| | Time-LLM | | PatchTST | |
|---|---|---|---|---|
| | p-Value: MAE | p-Value: MSE | p-Value: MAE | p-Value: MSE |
| ETTh1→ETTh2 | 0.050* | 0.065 | 0.045* | 0.058 |
| ETTh1→ETTm2 | 0.048* | 0.067 | 0.038* | 0.045 |
| ETTh2→TTh1 | 0.074 | 0.062 | 0.075 | 0.065 |
| ETTh2→ETTh1 | 0.050* | 0.074 | 0.043* | 0.040* |

Table 11: **Significance test on long-term forecasting**

| | Time-LLM | | PatchTST | |
|---|---|---|---|---|
| | p-Value: MAE | p-Value: MSE | p-Value:MAE | p-Value: MSE |
| ETTh1 | 0.0003* | 0.00025* | 0.05* | 0.05* |
| ETTh2 | 0.0001* | 0.0002* | 0.065 | 0.03* |
| Traffic | 0.075 | 0.063 | 0.05* | 0.03* |
| ILI | 0.055 | 0.072 | 0.06 | 0.0355* |

### A.4.6 SIGNIFICANCE TEST

To show the significance, we further conduct the significance tests by comparing ZeroTS against two best baselines Time-LLM and PatchTST on four selected tasks of both zero-shot learning and long-term forecasting. We implement it via t-test with scipy package of Python day and night, and demonstrate the empirical significance of ZeroTS where the p-value is denoted $p*$ when $p < 0.05$. The results are demonstrated as below Table. 10 and 11.

As observed in above two tables, on zero-shot forecasting, the results on 3 tasks out of 4 are of significance, while almost all results are of significance on selected long-term forecasting tasks (9 p-valuer results out of 16 are significant for MSE and MAE) where ZeroTS can exactly outperform Time-LLM and PatchTST on most scenarios by passing t-test. The minor scenarios our ZeroTS is inferior to these two baselines may be the dataset property and the fine-tune process on model adaptation where Transformer layers in our ZeroTS are not fine-tuned.

Thus, we still believe our work is novel insight that enables trade-off among data intelligence, model intelligence and efficiency issue. And our contributions lie in two aspects, i.e., not only the new perspective with RAG for data side adaptation and the ReinLLM coupling reinforcement learning with LLM to facilitate the optimization of a small representation model, but also the ultra-lightweight learnable parameters in our ZeroTS, providing possibility in edge computing and resource-limited services.

### A.4.7 MORE ANALYSIS ON HYPERPARAMETERS

The vital hyperparameters of our model can be five-fold illustrated in Table.6, number of retrieved auxiliary series $K$, representation dimensions of both target series and auxiliary series, cut-off coefficient $\rho$, step size for action selection $\eta$, as well as dimension of retrieved series.

**The most important one is the number of retrieved auxiliary time series, which is concerned with both effectiveness and efficiency.** The more series are retrieved, the more informativeness of prompts in LLM and increases the overall performance and generalization. But with more information retrieval, the computational costs on retrieval and fusion become higher. We can see that when $K$ arrives 6, our ZeroTS becomes pretty good performances and satisfies the trade-off.

**Regarding representation dimensions for both targeted and retrieved series,** we let them range from $\{8, 16, 24, 32, 64\}$ where it also requires the trade-off. It is known that larger dimensions usually account for more learning and fitting capacity, which corresponds to factual performance illustrated in Fig. 4. We thus take 64 as the final dimension as larger dimension will significantly increase the computational burden, e.g. 128.

**Considering the coefficient for cut-off retrieval** $\rho$. Such parameter determines which ranges of retrieved series can be obtained by our model and input into LLM as prompts. The larger $\rho$, sug-

Table 12: Hyperparameter setting on zero-shot prediction

| | ETTh1 → ETTh2 | ETTh1 → ETTm2 | ETTh2 → ETTh1 | ETTh2 → ETTm2 | ETTm1 → ETTh2 | ETTm1 → ETTm2 | ETTm2 → ETTh2 | ETTm2 → ETTm1 |
|---|---|---|---|---|---|---|---|---|
| Top-K retrieved series | 6 | 6 | 6 | 6 | 6 | 6 | 6 | 6 |
| Dimension of targeted TS representation | 64 | 64 | 64 | 64 | 64 | 64 | 64 | 64 |
| Dimension of retrieved TS representation | 64 | 64 | 64 | 64 | 64 | 64 | 64 | 64 |
| Cut-off coefficient $\rho$ | 0.3 | 0.3 | 0.3 | 0.3 | 0.4 | 0.4 | 0.4 | 0.4 |
| Step size for action selection $\eta$ | 0.05 | 0.05 | 0.05 | 0.05 | 0.05 | 0.05 | 0.05 | 0.05 |

gests more available series, and thus indicate more cues and evolution patterns obtained for LLM, yielding increasing model performances. To achieve the trade-off, we search the optimal one ranging from $\{0.2, 0.3, 0.4, 0.5\}$ and find 0.3 or 0.4 as optimal point for respective datasets. Actually, finer granularity can be explored but it requires more costs.

**Step size for action selection $\eta$.** It is the parameter for discretizing the coefficient $\alpha$ selection process in Eq.(7) thus for its adjustment. Since the coefficient $\alpha$ ranges from 0 to 1, the step stride is selected from $\{0.05, 0.1, 0.15, 0.2\}$ where larger step can lead to leap over optimal point while smaller step leads to inefficient dilemma. Finally, according experimental analysis and tests, we choose $\eta = 0.05$.

**Empirical results.** The hyperparameter adjustment on two selected scenarios are in Figure. 6 and Figure. 7. The final settings for zero-shot predictions are provided in the following Table. 12.

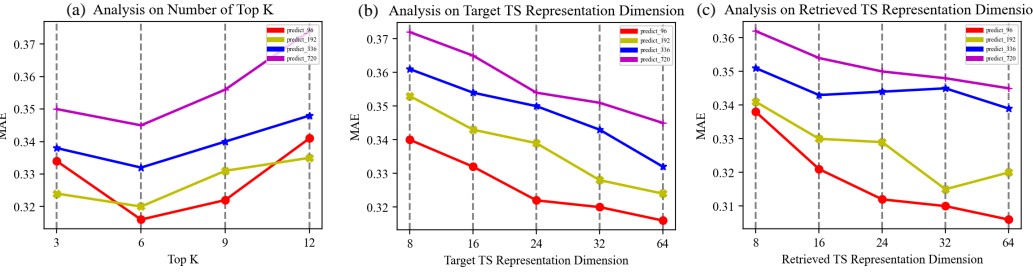

Figure 7: Hyperparameter analysis on ETTh1→ETTm2.

## A.5 LIMITATIONS AND FUTURE WORK.

Our work focuses on retrieving high-quality time series to enhance long-term and zero-shot series prediction with help of LLM, and enabling the data-model interactions with a small-scale learnable adapter. However, even effective and efficient, every coin has two sides. In this subsection, we discuss the potential limitations from three aspects. **Limitations.** First, due to limited computational resources and complexity constraints, we take GPT-2 as the backbone for prediction, which can be extendable to an update-to-date LLM such as Llama-3 when computational resources are sufficient. Second, reducing the complexity is a permanent topic for large-scale retrieval process and how to provide a selective process for filtering out irrelevant series. Third, as ZeroTS reveals less competitive on generalization from longer horizon to shorter horizon in Table. 1, it is suggested that the further improvement on how to transfer and adapt the coarse-grained data to fine-grained ones with series learning techniques. Therefore, our future work are also decomposed as three aspects on model architectures, high-efficient series retrieval and cross-granularity sequence learning. We will further point out a practical deployment solution to our ZeroTS.

**Future works.** **1) Higher version of LLM.** given that our work provides a scheme integrating reinforcement with time series, modeling more optimization options in series forecasting as discrete actions in reinforcement architecture, and it is possible to utilize higher version of GPT to achieve more accurate performances if time and resources are available. Then our system can support for more diverse and challenging scenarios such as out-of-distribution setting, cross-domain settings.

**2) Potential solution to more efficient retrieval.** A potential alternative towards the retrieval process can be pre-clustering the series prototypes, namely prototype-based retrieval. When the auxiliary series base constructed, we can first pre-cluster all the series into clustering as where one prototype represents a corresponding clustering. When a new series is added into the base, it will be assigned with one clustering and corresponding prototype can be updated. Given a targeted series, we can retrieve auxiliary series by matching target one to prototypes and reduce the complexity from $\rho^2 M$ to linear to M*log(N), where $N$ is the number of total series in auxiliary base, $M$ is the number of clustering of total series. **3) Overcoming long-short term transfer.** We speculate the reason behind this phenomenon is the changes of forecasting granularity. On overcoming such challenge, we can design a multi-resolution prediction scheme in training period, and enable the model to aware of the multi-grained evolution patterns and interpolation in the sequential elements with new learning objective. Then when the transfer is implemented, the coarse-to-fine pattern can be transferred from known granularity to unknown ones. We can also devise a temporal interpolation strategy and sequence augmentation to imitate series in fine-grained resolution, thus contributing to the adaptation on following transferred domain.

**4) Practical deployment of ZeroTS.** Furthermore, implementing the AI solutions in an efficient database framework is exciting (Ding et al., 2018). Since ZeroTS is an effective zero-shot forecasting foundation model, the practicality of its deployment is of importance. Considering the Time-Series oriented ZeroTS, especially with external time-series dataset by TS-RAG, we introduce an open-sourced temporal database named TD-engine (Taosdata, 2020), which is an industrial-grade database tailored for time-series and dynamic streaming data, to support high-performance, distributed IoT, industrial big data platform. Due to its open-sourced property and effciency on temporal data processing and indexing, it is expected to deploy our solution and algorithm on the such grant platform and we also believe the TD-engine can help our solution become more efficient and effective.

---

**Algorithm 2** HNSW Algorithm Pseudocode

```
 1: procedure BUILDHNSW(data, M, L, efConstruction)
 2:     for all vectors in data do
 3:         levels ← CalculateLevels(L, vector)
 4:         for all level in levels do
 5:             if level is not in graph then
 6:                 AddLevelToGraph(graph, level)
 7:             end if
 8:             AddVectorToLevel(graph[level], vector, M)
 9:             ConnectVectorWithNeighbors(graph[level], vector, efConstruction, M)
10:         end for
11:     end for
12: end procedure
13: procedure SEARCHHNSW(graph, query, M, efSearch)
14:     current_level ← max_level
15:     current_node ← ChooseStartNode(graph[current_level])
16:     results ← empty_list()
17:     visited ← empty_set()
18:     while current_level ≥ 0 do
19:         candidates ← FindNeighbors(graph[current_level], current_node, query, efSearch)
20:         candidates ← SortByDistance(candidates, query)
21:         for all candidate in candidates do
22:             if candidate is not in visited then
23:                 visited.add(candidate)
24:                 if length(results) < M then
25:                     results.append(candidate)
26:                 else
27:                     break
28:                 end if
29:             end if
30:         end for
31:         if length(results) ≥ M then
32:             break
33:         end if
34:         current_node ← BestNeighbor(results, query)
35:         current_level ← current_level - 1
36:     end while
37:     return results
38: end procedure
```

Figure 8: Pseudocode of HNSW

