# OpenReview forum: "ZeroTS: Zero-shot time series prediction via multi-party data-model interaction"
_ICLR.cc/2025/Conference — ICLR 2025 Conference Withdrawn Submission_

### Official Review · Reviewer_YKdy · 2024-10-28

**Soundness:** 2
**Presentation:** 1
**Contribution:** 2
**Rating:** 3
**Confidence:** 4

**Summary:**

This paper investigates the use of large language models (LLMs) for zero-shot time series forecasting. To address scalability and flexibility, it proposes a solution using data-model interaction, including a TS-RAG data prompt and a reinforcement learning framework. Extensive experiments on zero-shot and long-term forecasting validate the effectiveness of the approach.

**Strengths:**

This paper proposes a TS-RAG data solution and a reinforcement learning framework. Extensive experiments demonstrate its effectiveness, and the code is provided.

**Weaknesses:**

1. The writing can be significantly improved. "TIME SERIES PREDICTION,"should be changed to "forecasting." ; Be better to use more common "TSF" instead of "TS" as the abbreviation for time series forecasting. Additionally, some expressions, like "While existing models, particularly large language models (LLMs) tailored for TS," are confusing. As far as I know, no LLMs are specifically tailored for time series, making this statement difficult to understand.

2. The motivation is not very clear and lacking a discussion of related work and comparision with key baselines. Many foundational time series models [1-5] have already been proposed, demonstrating strong zero-shot TSF capabilities with fewer parameters than LLMs. Furthermore, studies [6] have validated that LLMs are not useful for TSF, i.e. not  better than random initialization. In this context, the paper aims to use LLMs for zero-shot TSF, which raises questions about the motivation, both in terms of effectiveness and efficiency. The authors should discuss and Explicitly compare with foundational time series models; Directly address the findings from [6] about LLM effectiveness for TSF and explain how your approach differs or overcomes these limitations.

3. In addition, the zero-shot forecasting setting in this paper involves transferring from domain A to domain B. I acknowledge that this is also referred to as zero-shot in some literature. However, I am curious whether the paper could consider the more practical zero-shot setting used in [1-5], more "out-of-the-box", already well-supported by foundational time series models. Please expand evaluations to include this more practical zero-shot setting.

4. Allowing time series from different domains to serve as prompt may cause potential information leakage. Such as use data from other domains but in the future time period.

[1]Liu, Yong, et al. "Timer: Transformers for time series analysis at scale." ICML 2024.

[2] Liu, Juncheng, et al. "UniTST: Effectively Modeling Inter-Series and Intra-Series Dependencies for Multivariate Time Series Forecasting." ICML 2024.

[3] Goswami, Mononito, et al. "Moment: A family of open time-series foundation models." ICML2024.

[4] Ansari, Abdul Fatir, et al. "Chronos: Learning the language of time series." arXiv preprint arXiv:2403.07815 (2024).

[5] Rasul, Kashif, et al. "Lag-llama: Towards foundation models for time series forecasting." R0-FoMo: Robustness of Few-shot and Zero-shot Learning in Large Foundation Models. 2023.

[6] Tan, Mingtian, et al. "Are language models actually useful for time series forecasting?." arXiv preprint arXiv:2406.16964 (2024). Neurips 2024.

**Questions:**

Please check limitations

---

> ### Author Response · Authors · 2024-11-20
> **Rebuttal for Reviewer YKdy (1/3)**
>
> Dear Reviewer YKdy,
>
>   Thanks for your valuable time and comments on our research, we will heavily improve our manuscript to satisfy the high standards of ICLR community. The detailed responses can be found as below.
>
> **W1. Expression of TSF and LLMs for TSF.** Thank you for your suggestions and we will modify our words into common expressions, e.g., ‘time series forecasting’ and corresponding ‘TSF’ for abbreviation.
>
> **Literature and Classification of LLM for Time Series (LLM4TS).**
> Actually, some recent solutions have been proposed that modify LLM to be tailored for Time Series, including GPT4TS (NeurIPS 23) [7], Time-LLM (ICLR24) [8], and LLMTime (LLM for zero-shot time-series forecasters, NeurIPS23) [9] as well as LeRet (take both language and series as inputs for multi-modal fusion, IJCAI 24) [10]. They can be divided into two-fold,
>
> - Fine-tune some layers of LLM and adapt the LLM to numerical time-series (with major component of LLM frozen), e.g., [7] and [8].
> - Transform the input numerical time-series into word characters that can be identified and processed by LLM and LLM is not fine-tuned. e.g., [9], [10].
>
> In addition, there is a subsection named **LLM4TS and RAG** in our Related Work of main text, where most LLM4TS models have been discussed. Please refer to this subsection for more details.
>
> All above works have investigated how to develop LLM-based time series forecaster to obtain more generalization capacity. As the reviewer mentioned in Weakness 2, Ref [5] is also a foundational Language Model for time series forecasting, and all Ref.[1-5] utilize the Transformer as backbone module while actually the basic module of GPT-2 (only 700M parameters) in our work is also Transformer. Thus, it is rational to exploit GPT-2 to implement the zero-shot TSF task and it should not be blamed as well. More discussions on efficiency and effectiveness can be found in next subsection of our response.
>
> **W2. Motivation and differences against existing LLM model and other foundational models**
> The questions are two-fold. First, it is concerning the motivation why not utilize the foundational time-series models. Second, it is to argue against LLM is not suitable for TSF. We respond to above questions as below. We have explicitly discussed different models and compare our solution with both conventional foundation model and recent LLM in both practical and empirical manners.
>
> **W2-1. Clarification on efficiency issue and boundary of conventional foundational model and LLM.** Actually, the architectures (backbones) for forecasting in literature [1-5] are all different variants of transformer, i.e., Timer, UniTST, MOMENT, Chronons [1-4] are transformer-based and Ref.[5] is a recent LlaMA-based model. Noted that these transformer architectures in [1-5] require training from scratch and the parameter scales of abovementioned models range from Million level to Billion level according the scale of datasets. In contrast, our ZeroTS is a rather lightweight model compared to them, which inherits the architecture of GPT-2 without training or fine-tuning such GPT. Our model is with 50M Trainable Param with 700M of GPT-2 for inference, which is lighter than above methods. Furthermore, in fact, **there is not an explicit boundary between the conventional foundation models and LLM**, the full name of GPT-2 is generative pretraining Transformer at version 2, where the inner structure is also transformer. Thus, the reviewer should not blame on the utilization of transformer-based GPT-2 as the provided references are all transformer-based. Empirically, to further demonstrate the performance difference between LLM and conventional foundation models, we take a more recent Llama-3 and a conventional foundation model PatchTST as instances for experimental tests, and provide performance comparisons with two settings, i.e., RAG coupled with Llama-3, and RAG coupled with PatchTST, as well as vanilla PatchTST. We report the results of MSE on both zero-shot and long-term predictions as below.
>
> |       MSE     | RAG+Llama-3 | RAG+PatchTST | RAG+GPT2 (ZeroTS) | PatchTST |
> |:-----------------------:|:-----------:|:------------:|:------------:|:------------:|
> | ETTh1$\rightarrow$ETTh2 |    0.345    |    0.361    |   0.350  |   0.380   |
> | ETTh1$\rightarrow$ETTm2 |    0.274   |    0.286    |   0.272  |   0.314  |
> |          ETTh1        |    0.402    |    0.410   |   0.403  |   0.413  |
> |          ETTh2        |    0.312    |    0.319    |   0.315  |   0.330   |

---

> ### Author Response · Authors · 2024-11-20
> **Rebuttal for Reviewer YKdy (2/3)**
>
> **Analysis on empirical results.** From above results, two major conclusions can be obtained. In our zero-shot and long-term forecasting, when the LLM is replaced with **conventional model PatchTST**, the performances have significantly dropped ranging from 1.27\% to 5.15\% where zero-shot scenarios are more sensitive to backbones, and 5\% performance differences should not be neglected. For different LLMs, when GPT-2 is replaced with Llama-3, the performances are with little fluctuations ranging from 0.2\% to 1.4\% which is acceptable. For efficiency, Llama-3 is equipped approximately 8 Billion para. for inference while GPT-2 only has 700M para. with 50M as learnable parameters. Therefore, our conclusion is LLM is of necessity and a lightweight LLM such as GPT-2 will be sufficient for our zero-shot tasks.
>
> **W2-2. Novelty and perspectives.** Actually, instead of fine-tuning layers of LLM in GPT4TS (NeurIPS’23) [7] and Time-LLM (ICLR’24) [8], our ZeroTS detours fine-tune on LLM and takes LLM as a tool and trains a small model to activate the power of LLM, leading to output of LLM towards facts. This is more efficient and effective when implementing zero-shot task. We argue that the flexibility lies in the updatable dynamic auxiliary time-series base and the lightweight learnable representation for activating LLM.
>
> **W2-3. Discussions on how to adapt models to various scenarios.**
>
> In Ref.[6], it has conducted simple ablations on ordinary time-series forecasting and some few-shot scenarios. But it has also mentioned that ‘our goal is not to suggest that LLMs have no place in time series analysis. To do so would likely prove to be a shortsighted claim…more focus to the exciting tasks could be unlocked by LLMs at the interface of time series and language such as time series reasoning, or social understanding’. The core of this research is to inspire more exciting research of exploiting LLM for more challenging, generalized and novel tasks. Actually, in our research, we consider LLM is a tool for extending the power of artificial intelligence and data, **which is inherently under such guidance**. We believe whether LLM is appropriate for the task is determined by the property of task itself. Let us provide some discussions. Usual time-series such as Electricity and traffics have strong periodicity and few fluctuations. In this context, directly forecasting these series or predicting time series in a specific domain, LLM is exactly not a good choice.
> However, when implementing the zero-shot forecasting, i.e., transferring the model from domain A to B, or directly forecasting series in a brand-new domain without fine-tuning, LLM can be a good helper. In fact, LLM is equipped with relatively more parameters which is trained by numerous textualized language data, which is considered to be a good tool for assimilating the open-world knowledge and intelligence that is outside of the pure data. If we take it as a tool without fine-tune, our inference is still lightweight free from efficiency issue, as shown in Table. 5 in our manuscript.
> To this end, we argue that LLM is suitable for generalization scenarios with high demands on sequential semantic understanding and data fusion, 1) **Complex and challenging scenarios** such as zero-shot or few-shot scenarios that require supplementing external temporal patterns from the open world. 2) **Require multi-modal inputs**, especially the language data for supplementing the status of exogeneous factors. To resolve the zero-shot challenge, we choose GPT-2 as our backbone, which is with fewer parameters than even PatchTST (a claimed ordinary series learning model).
>
> Thank you again for raising above questions and we will carefully incorporate these additional discussions into our revised manuscript. Noted that although some of the provided references are in the same period with ICLR 2025, we have also discussed them in our responses to enhance the quality and comprehensiveness of our research and cited them appropriately.
>
> [7] One Fits All: Power General Time Series Analysis by Pretrained LM, NeurIPS 2023.
>
> [8] TIME-LLM: TIME SERIES FORECASTING BY REPROGRAMMING LARGE LANGUAGE MODELS, ICLR 2024.
>
> [9] Large Language Models Are Zero-Shot Time Series Forecasters, NeurIPS 2023.
>
> [10] LeRet: Language-empowered retentive network for time series forecasting. IJCAI 2024.

---

> ### Author Response · Authors · 2024-11-20
> **Rebuttal for Reviewer YKdy (3/3)**
>
> **W3. More out-of-the-box scenarios.** Thanks for your valuable suggestions. To further verify the effectiveness, we have expanded experiments to a more general setting used in [1-5], i.e., with no particular task-specific exemplars for tuning. In our ZeroTS, we utilize the non-training GPT-2 with RAG (retrieval strategy TS-RAG) for implementation. ZeroTS has degenerated into a pure RAG model without learnable parameters and strategies. We also take Llama-3, PatchTST as comparisons with fair settings. Three typical datasets, i.e., ETTh1, ETTm1, and Traffic are selected to illustrate the out-of-the-box prediction results of our non-trainable version of ZeroTS.
> |             | RAG+Llama-3 | RAG+PatchTST | RAG+GPT-2(ZeroTS) |
> |:--------------------------:|:-----------:|:------------:|:----------------:|
> | Full zero-shot on ETTh1 |   0.442    |     0.485     |       0.457      |
> | Full zero-shot on ETTm1|   0.440    |     0.465     |       0.443      |
> | Full zero-shot on Traffic |    0.456    |     0.504    |       0.472      |
>
> Actually, to this end, our design of RAG+GPT-2 can outperform conventional model PatchTST, and achieve comparable performances with RAG+Llama-3 but with sacrificing large-scale computational loads. Therefore, we can conclude that the choice of GPT-2 is fine and superior to others regarding efficiency and effectiveness trade-off.
>
> **W4. Clarification on potential information leakage.** We have considered the issue of information leakage into consideration with the following two designs.
> 1)	First, to avoid the information leakage of LLM, and fairly investigate the effectiveness of our idea, we exploit GPT-2 (released in 2019) instead utilizing a most recent model as backbone, and the testing/inference datasets are all released after 2019. That’s to say, the LLM has not seen the training/testing set when experiencing the pre-training.
> 2)	Second, to avoid the information leakage of RAG, we utilize the disjoint datasets in retrieval ones and inference ones and ensure the ordered timestamps in the retrieval. More specifically, in our testing stage, to ensure fair testing and imitate the real-status, we filter the available series with meta information (‘timestamps’) and directly mask the ‘future observations’ corresponding to this targeted series as unseen ones (the remaining $1-\rho$). To this end, we can ensure our solution undergo a fair and trustworthy test. And in deployed real-world applications, if we are expected to predict future steps, actually only previous events and observations are available as no one can reach the future beyond the present.
>
> With above well-designed strategies, we can declare that our solution will not encounter the information leakage in both real applications and testing periods. Thanks for raising this question and we will emphasize these designs avoiding the information leakage in our method and experiments as clarifications.
>
> Finally, we would like to thank you for raising these questions and discussions for us, and we have carefully responded to your concerns and questions. More discussions on LLM, and conventional models, as well as new empirical results of out-of-the-box scenarios will be incorporated into our manuscript. We hope you can support our work at the next stage.
>
> Best,
>
> Authors of Paper 705

---

> > ### Author Response · Authors · 2024-11-23
> > **Further discussion**
> >
> > Dear Reviewer YKdy,
> >
> > Thank you very much for devoting time on reviewing our manuscript. As the author-reviewer discussion process is ending soon, we are wondering whether our responses have well addressed your concerns. If you have any further questions regarding our manuscript, please let us know and we are glad here to provide further discussion and clarification to improve the quality of this manuscript. Thanks.
> >
> > Authors of Paper 705

---

> ### Author Response · Authors · 2024-11-28
> **A brief rebuttal for Reviewer YKdy**
>
> Dear Reviewer YKdy,
>
> Thanks for your valuable time and comments on our research.  For your convenience, we have summarized the core responses and changes according to your valuable reviews. You can kindly find it below.
>
> **1) Expression of TSF and LLMs for TSF.** We have modified our words into common expressions, i.e., ‘time series forecasting’ and corresponding ‘TSF’ for abbreviation.
>
> **2) Literature and Classification of LLM for Time Series (LLM4TS).** LLM4TS can be divided into two folds.
>
> -	**Fine-tune some layers of LLM** and adapt the LLM to numerical time-series (with major component of LLM frozen), e.g., GPT4TS (NeurIPS 23), Time-LLM (ICLR24).
>
> -	**Transform the input numerical time-series into word characters** that can be identified and processed by LLM and LLM is not fine-tuned. e.g., LLMTime (LLM for zero-shot time-series forecasters, NeurIPS23) [9] as well as LeRet (take both language and series as inputs for multi-modal fusion, IJCAI 24).
>
> Actually, there is a subsection named LLM4TS and RAG in our Related Work of main text, where most LLM4TS models have been discussed.
>
> **3) Motivation and differences against existing LLM model and other foundational models.**
>
> GPT is **inherently a transformer-based model for language representation extractor**. Many foundational models **are not necessarily** equipped with fewer parameters than GPT-2. There is also **not an explicit boundary between the conventional foundation models and LLM**. We also perform **additional experiments for comparison** among (1) RAG+recent new LLM (Llama-3), (2) RAG + foundational model (PatchTST), and (3) RAG + GPT-2 (ZeroTS) to demonstrate the efficiency and effectiveness of ZeroTS.
>
> Instead of fine-tuning layers of LLM in GPT4TS (NeurIPS’23) and Time-LLM (ICLR’24), ZeroTS detours fine-tune on LLM and **takes LLM as a tool** and **trains a small model** to activate the power of LLM, leading to output of LLM towards facts. This is more efficient, effective and flexible when it comes to zero-shot task. The **flexibility** lies in the updatable dynamic auxiliary time-series base and the lightweight learnable representation for activating LLM.
>
> **4) Argue against LLM4TS and how to adapt models to various scenarios.** The core of this research is to **inspire more exciting research of exploiting LLM for more challenging, generalized and novel tasks**. In our research, we consider LLM is a tool for extending the power of artificial intelligence and data, which is **inherently under such guidance**. We believe whether LLM is appropriate for the task is determined by the property of task itself.
>
> We consider LLM is suitable for scenarios with high demands on sequential semantic understanding and data fusion, they are two-fold. **1) Complex and challenging scenarios** such as zero-shot or few-shot tasks requiring supplementing temporal patterns from the open world. **2) Multi-modal inputs**, especially the language data for supplementing the exogeneous factors.
>
>
> **5) More out-of-the-box scenarios.** We have conducted **additional experiments on full zero-shot tasks**, i.e., out-of-the-box scenarios on three settings (RAG+Llama-3/RAG+PatchTST/RAG+GPT-2(ZeroTS)) and three datasets (ETTh1/ETTm1/Traffic). The results can also verify the effectiveness and rationality of our proposed ZeroTS.
>
> **6) Clarification on issue of Information leakage.**
>
> -	The LLM model does not leak the data in previous training process.
> -	The disjoint datasets for retrieval ones and inference ones, and the ordered timestamps in the retrieval set avoids the leakage in temporal dimension.
> -	In deployed real-world applications, regarding predicting future steps, actually only previous events and observations are available as no one can reach the future beyond the present.
>
> Thank you very much for spending time on reviewing our manuscript. We have revised our manuscript and carefully incorporated these experimental results and discussions into our manuscript.  We believe the overall quality of our manuscript has been significantly improved **according to your valuable comments and suggestions**. Now, we are wondering whether our responses and detailed revisions have addressed your concerns. If so, please kindly re-consider your rate of our research. If you have further questions, please feel free to let us know and we are glad to provide further discussions and explanations. We are looking forward to your support! Thank you!
>
>
> Best,
>
> Authors of Paper 705

---

### Official Review · Reviewer_uNhX · 2024-10-29

**Soundness:** 2
**Presentation:** 2
**Contribution:** 2
**Rating:** 3
**Confidence:** 3

**Summary:**

This paper addresses the challenge of zero-shot time series prediction by introducing TS-RAG, a retrieval-based model that leverages an external database to enhance prediction performance. The authors begin by constructing a key-value database for retrieval-augmented generation  and subsequently design a retrieval scheme known as HSNSW. Additionally, they propose a reinforcement learning framework that incorporates both a policy and a value network to facilitate interaction between the data and the model, ultimately refining the output.

**Strengths:**

1. The problem addressed in this paper is significant, and the approach of exploring RAG in the context of time series prediction is novel.
2. The experimental results demonstrate a substantial improvement over other baseline methods.

**Weaknesses:**

1. The presentation of the paper requires refinement. For instance, the subcaptions in Figure 1 are encoded within the image and are too small to read. Additionally, the light green font used for references is difficult to see.

2. While the idea of incorporating external knowledge to enhance time series forecasting is noteworthy, it is not entirely new. Moreover, the paper introduces two key component, TS-RAG and LR for adaptation, but these elements are presented in isolation, which makes the overall contribution feel fragmented.

3. The choice of retrieval database is crucial for the practical application of this algorithm, yet this aspect is not addressed in the paper. The potential impact of varying the database remains unclear, which undermines the technical soundness of the approach.

4. The organization and writing of the paper need improvement. For example, $ W_c $ is introduced in Equation (1), but it is not referenced or explained in the following content, leading to confusion. Many notations and model details are unclear. For instance, what are HSNSW and HNSW, and are there any references that explain these concepts?

**Questions:**

1. You utilize three retrieval databases that do not overlap with the target datasets. How were these datasets chosen—was it based on expert knowledge? Is there a necessity for any relationship between the retrieval database and the inference dataset, such as sharing the same domain? Additionally, are there specific requirements regarding the scale of the databases?

2. Could you provide a clearer explanation of ReinLLM? What constitutes the action space of the policy network—does it only include kernel size? Besides, the whole adapter functions an additional component separate from the LLM. If so, why not simply use an additional network specifically for fine-tuning instead?

3. Recent studies on time series foundation models have focused on training across various domains of time series datasets. In this context, I have concerns about the impact of your work. What would happen if you trained a cross-domain model using your retrieval database in conjunction with the training dataset, or even just used the training dataset for fine-tuning?

I would consider increase my score if you can address my concerns about the choice of database and the necessity of your RL adaptor.

---

> ### Author Response · Authors · 2024-11-20
> **Rebuttal for Reviewer uNhX (1/4)**
>
> Dear Reviewer uNhX,
>
> Thanks for your constructive comments and valuable time on our manuscript. We hope we can improve the quality of this research altogether and make our best efforts to satisfy the high standards of ICLR community. The detailed responses can be found as below.
>
> **W1. Text color of references.** Thanks for raising this issue and we will change the reference color to blue and make it easier to read.
>
> **W2. Idea of ZeroTS and Coherence of two modules, TS-RAG and ReinLLM.**
> In this research, we systematically resolve the challenge of an emerging task of zero-shot time-series prediction via **multi-part data-model interactions**. Compared to ordinary series prediction, zero-shot task requires more flexibility and less training data where data is restricted. Specifically, the **new perspective** lies in devising a lightweight learnable model to maximumly active the power of LLM and sufficiently assimilate the cues from external but updatable information, enabling both flexibility and efficiency. From data side, we exploit a RAG strategy to dynamically retrieve similar sequences as additional cues for LLM prompting. Unlike conventional knowledge graph driven requiring complex graph operations, this allows the external knowledge can be updated anywhere and anytime in a flexible manner. From model side, we take errors between LLM output and ground-truth to allow our learnable parameters to dynamically correct the series representations to approach real-world facts. The efficiency of ZeroTS lies in its limited learnable parameters without any fine-tuning on LLM, where the adopted LLM, i.e., GPT-2 is also lightweight. We believe the multi-party interaction perspective is novel to this field, and the techniques of efficient series retrieval, ReinLLM, a reinforcement scheme enabling LLM feedback back-propagate to small model adjustment are soundness.
>
> **Coherence between two models.** Actually, to realize the data-model interactions, enabling our model to extract external series and how to fuse the external series with a learning objective is a natural intuition in our task. To this end, we first propose TS-RAG to realize an efficient retrieval from external series dataset. To fuse retrieved series and maximumly activate the power of LLM with RAG in time series, we devise ReinLLM adapter, it receives retrieved series as input and interconnects with LLM to derive the results, then the errors backpropagate to learnable representations on both targets and prompts, thus guiding the fusion of retrieved time series and activating the maximal power of LLM. The overview and data flow illustration are in Fig.2 of our manuscript, and we believe they are in a consistent architecture, and jointly implement the data-model interactions from data and model sides, activating the output of LLM towards the ground-truth.
>
>
> **W3&Q1. Choice and impacts of retrieval auxiliary time-series set.**
>
> **How to obtain appropriate series in ZeroTS.** We acknowledge that the Reviewer is expected to know how to select the retrieved series. Actually, in our retrieval process, our solution is in a data-driven perspective **without necessity** for deriving relationship between retrieval database and the inference dataset. In detail, we retrieve the auxiliary time-series by first matching the meta information including domain category, timestamps, and calculating the series-level similarity for filtering of most relevant ones. With the matching scheme of meta information, the scope of auxiliary time-series for retrieval has been limited, thus we do not need other specific strategies to choose the set.
>
> **Investigation of impacts of selected series.** Regarding the impacts of selected auxiliary series, actually, we utilize a hyperparameter of $\rho$ in a $\rho-\rho$ cut-off scheme to control the available scales of auxiliary series. The investigation and performances with different $\rho$ can be found in Figure 6 (a)-(b) in our main text. The more series are available for retrieval, the computational cost will increase and potentially more noise (possibility of involving more unrelated series is increasing) is involved. Thus, we choose $\rho=0.3$ in an empirical manner.

---

> ### Author Response · Authors · 2024-11-20
> **Rebuttal for Reviewer uNhX (2/4)**
>
> **More discussion about the dataset selection, relationship and scale of database.** According to above analysis, in our work, we couple the meta-information matching with similarity-based retrieval to search the most related series to target one, where meta information is used to conduct coarse-grained filter and similarity computation conducts finer-grained selection, as more similar sequence may provide more accurate cues for target prediction. Therefore, we **do not specify** the relationships between targets and auxiliary bases, and let it select in a natural manner. Regarding the scales, we think it can be a trade-off among involved noise (more series can potentially involve unrelated ones even though share some similarity), efficiency (more series indicate more computations) and performance, and the final proportion ($\%$, 0.3\% in this work) of whole base can be determined by the empirical experiments.
>
> **More insights from your comments.** For future research, we can pre-prepare the related available auxiliary domains and adopt a hierarchical series-level base, where hierarchical series can be composed of series from less related to most related. Given a targeted series of numerical weather in a specific city, we can select the population, traffics, previous weather and nearest weather in the same city to construct the auxiliary base, where these sets are from coarse-grained relations to fine-grained relations. On retrieval, we can implement a prototype-based series clustering where each clustering has a corresponding prototype. Then we can perform inter-clustering search by matching the targeted series with each prototype and then conduct intra-clustering search for fine-grained match, which reduces the computations via pre-computed clustering prototype and hierarchical matching scheme. Thanks for your suggestions and we will incorporate these discussions in our manuscript as well as the in the Appendix.
>
> **W4. Details provided in Eq.(1) and Algorithm 2.**
> For the weights $W_c$ and $W’_c$, they are learnable weights as an AutoEncoder, which transfers the retrieved time-series into a compressed representation for further processing. The explanation on this scheme lies in Line 229-231 on Page 5 of our main text.
>
> For Hierarchical Navigable Small World (HNSW), its core idea is to organize different samples into a multi-layered graph, where each layer is a network of small world, with upper layer providing a quick search and the lower layer providing a finer granular search [1,2]. We then let it modify into a series-level retrieval process, name this mechanism as Hierarchical Series-level Navigable Small World (HSNSW). Actually, we implement it by the Faiss which is an open-sourced package developed by Facebook. We are sorry that ‘Fasiss’ in manuscript should be corrected as **‘Faiss’** [3]. We have provided corresponding references to both theory of HNSW [1,2] and implementation of them [3]. We hope these explanations can help reviewer better understand our work. Thanks for your questions and suggestions, and they are valuable for quality promotion.
>
> [1] Malkov Y A, Yashunin D A. Efficient and robust approximate nearest neighbor search using hierarchical navigable small world graphs[J]. IEEE transactions on pattern analysis and machine intelligence, 2018, 42(4): 824-836.
>
> [2] Zhong Y. Efficient Implementation of Hierarchical Navigable Small World Similarity Matching Algorithm[M]. University of California, San Diego, 2020.
>
> [3] Johnson J, Douze M, Jégou H. Billion-scale similarity search with GPUs[J]. IEEE Transactions on Big Data, 2019, 7(3): 535-547.

---

> ### Author Response · Authors · 2024-11-20
> **Rebuttal for Reviewer uNhX (3/4)**
>
> **Q2. Detailed explanation of ReinLLM.**
>
> **Q2-1. Details on ReinLLM.** ReinLLM is an adapter connecting auxiliary time-series retrieval and predictor of LLM, which receives retrieved series and errors from LLM, and then backpropagate errors to learnable representations on both targets and prompts, thus guiding the fusion of retrieved time series and activating the maximal power of LLM. As shown in Fig.3, our policy network consists of two major components, i.e., temporal kernel selection for target series representation and auxiliary series fusion coefficient selection for time-series fusion. Thus, our action space **do not only include kernel size**, but includes two important hyperparameter selection for respective targeted and auxiliary series representations.
>
> **Q2-2. Reasons for not simply fine-tuning LLM instead.** Actually, when directly fine-tune on LLM, even though with a small model, it will encounter three challenges, 1) efficiency and time cost issue, 2) unavailable data for LLM fine-tune in zero-shot scenarios. 3) Flexibility issue for adapting new data. Specifically, the limited new data cannot enable the changes of parameters of LLM. In contrast, we propose our ReinLLM in ZeroTS that may be more suitable for Zero-shot time-series prediction. The adapter actually serves as a separated lightweight learnable module to fine-tune the LLM without changes of parameters of LLM. This mechanism interconnects external series and LLM with small scale learnable weights, thus also enjoying the flexibility and efficiency for generalization to a broader plug-and-play external knowledge.
>
> **Q2-3. Reasons for reinforcement.** The property and superiority of reinforcement learning is to implement active interactions between model and environments, and then dynamically optimize multi-end neural network actions. RL simulates agents in the world to take actions interacting with environment and changing the state to achieve optimal rewards. In our work, we have both external data and LLM-oriented model as environments, while we have representations of both targeted series and fused auxiliary series for optimization. To this end, it is more appropriate to model our tasks with reinforcement architecture for multiple-end optimization, where the auxiliary series and LLM can be both considered as environments, critical parameters in NNs can be viewed as actions, the representations are states, negatived error as reward, thus small learnable ReinLLM (adaptor) can connect the external data-side insights with LLM side wisdom. This scheme especially maximumly exploits power and error feedback from LLM (as reward) to optimize previous representations.
>
> Thanks for raising this question, and we will enhance our motivation and provide more discussions to make it clearer.
>
> **Q4. Distinguish ZeroTS with other cross-domain learning and multi-task learning work.**
>
> Thanks for raising this question. It is a good idea to train cross-domain model using data from different domains, and it has been verified for its effectiveness in pioneering literature [4],[5]. We can name it as unified learning or continuous multi-task learning. For [4], it falls into the idea of **training a cross-domain model using multi-sourced/domain data** and it exactly demonstrates the improvement and zero-shot capacity. For **training and fine-tune scheme**, [5] devises a task-level continuous learning to iteratively fine-tune the new task with partial neuron stable, which also provides additional capacity in zero-shot and cold-start scenarios for dynamic learning. Even so, we argue that the unified model cannot adapt all scenarios and lack flexibility to continuous update [4] while continuous learning lacks the flexibility to retrieve any external knowledge and requires additional fine-tune parameters, limiting the zero-shot learning capacity. We think the key point of this research line is to decouple the invariant and variable parts, and transfer the invariances to new domain (task) while assimilate changes from environments.
>
> **A toy example test on cross-domain learning.** A toy experiment on NYC (with crowd in/out and taxi pick/drop four domain series) is obtained, when training on other 3 tasks, and the 4th task with only 25\% samples, the cross-domain learning can improve performance of MAPE by 5\% from best baseline 0.473 (PromptST) to multi-task learning setting 0.450 (cross-domain). Even so, cross-domain learning still suffers two critical issues **1) only correlated tasks and domains can be utilized to reinforce each other and blindly involving various tasks can lead to model suffering noise and distractions for main predictions**, **2) the unified model may not accommodate patterns from all tasks that limits the extensibility**.

---

> > ### Author Response · Authors · 2024-11-20
> > **Rebuttal for Reviewer uNhX (4/4)**
> >
> > To this end, we argue that ZeroTS is a totally different perspective to address zero-shot and generalization task, which detours the large model fine-tune, and is with more flexibility and efficiency without considering task-level or domain-level similarity. Our solution allows any time-series to be retrieved, and incorporated into the model dynamic update, utilizes the additional temporal evolution cues (mostly similar to target one) to prompt the LLM for output prediction. ZeroTS can simultaneously exploit the power of external knowledge in a plug-and-play manner and informative semantics and regularity in LLM in a lightweight manner.
> >
> > Thanks for this question and inspiring us deeper insights on differences between our ZeroTS and cross-domain learning tasks. We will incorporate these discussions into our main manuscript or Appendix, especially the discussion on multi-task/cross-domain learning to raise and enhance our motivations. Thanks.
> >
> >
> > [4] Yuan Y, Ding J, Feng J, et al. Unist: a prompt-empowered universal model for urban spatio-temporal prediction[C]//Proceedings of the 30th ACM SIGKDD Conference on Knowledge Discovery and Data Mining. 2024: 4095-4106.
> >
> > [5] Yi Z, Zhou Z, Huang Q, et al. Get Rid of Task Isolation: A Continuous Multi-task Spatio-Temporal Learning Framework[J]. arXiv preprint arXiv:2410.10524, 2024.
> >
> > We want to sincerely thank the reviewers again for the valuable time on our work and we believe the insightful questions can exactly help promote quality of our research as well as this manuscript.
> >
> > Best,
> >
> > Authors of Paper 705

---

> > > ### Author Response · Authors · 2024-11-23
> > > **Further discussion**
> > >
> > > Dear Reviewer uNhX,
> > >
> > > Thank you very much for devoting time on reviewing our manuscript. As the author-reviewer discussion process is ending soon, we are wondering whether our responses have well addressed your concerns. If you have any further questions regarding our manuscript, please let us know and we are glad here to provide further discussion and clarification to improve the quality of this manuscript. Thanks.
> > >
> > > Authors of Paper 705

---

> > > > ### Comment · Reviewer_uNhX · 2024-11-25
> > > > **response**
> > > >
> > > > Thank you for your thoughtful response and the improvements made to the manuscript. I appreciate the authors' efforts in addressing my concerns. However, I remain unconvinced by the explanation regarding the need for specific selection or design of the auxiliary time-series database.
> > > >
> > > > As mentioned, you argue that "matching the meta information, including domain category, timestamps, and calculating the series-level similarity for filtering the most relevant ones" limits the scope of the auxiliary time-series for retrieval, thereby eliminating the need for additional selection strategies. However, this implies that the auxiliary database must include a broad range of domain categories and timestamps to ensure successful matching based on meta information.
> > > >
> > > > My concern is what happens when the database is small or contains only a limited number of domain categories and records. In such cases, if the auxiliary time-series bear little or no correlation with the inference time series, the matching process would likely be ineffective. Could you clarify how the approach accounts for such limitations in the auxiliary database, particularly when there is insufficient variation in the data or when matching results in a low similarity?

---

> ### Author Response · Authors · 2024-11-25
> **Further discussion on dataset selection and motivation of our design**
>
> The authors sincerely appreciate the reviewer's follow-up discussions. We provide further clarification.
> As we know, from a classical signal decomposition theory [6,7], time-series tends to be decomposed of different components, or a complex series can be the combinations of several simple components. Thus, we can take this theory to interpret our auxiliary data based solution.
> Our retrieval and fusion take a Top-K selection mechanism and then dynamically learn the fusion coefficients with reinforcement scheme. Our model doesn't require the intensive domain-related data but exploit the intuition in our Fig.1 that series share similar trends in the world can reinforce each other.  Even though series have limited direct similarity, their combinations may make sense.  In our solution, we can always find the Top-K series and dynamically combine them in a learnable manner and further **provide the cues of potential trends for output of the backbone language model**.  Of course, there must be unavoidable performance variations on dataset selection. Our intuition is that when the scale and diversity of external series set is increasing, the more patterns are included, but we **cannot numerate all time-series** in the world and finding additional series will also **lead to further unexpected complexity and burden on collection**. The only way we can do **is to eliminate the variation influences on external dataset selection and mine the potential relationships and combined patterns between target and retrieval ones with trade-off**. Given above, we believe our scheme have provided insights into the how combined/decomposed series facilitate the zero-shot prediction and can maximumly reduce the influences of dataset selection from both empirical (experimental  setting like meta information selection and similarity comparison) and theoretical support [6,7]. Thanks for your question again and we will incorporate our further discussion into our Appendix.
>
> --------------------------------------------------------------------------------------------------------------------------------------------------------------------------
> We have updated our manuscript with above discussion and potential clustering/prototype-based selection strategy **in our manuscript**. Please refer to the **new subsection A.1.5 (Discussion on  selection of auxiliary dataset)**. Thanks.
>
> [6] Akansu A N, Liu Y. On-signal decomposition techniques[J]. Optical Engineering, 1991, 30(7): 912-920.
>
> [7] Akansu A N, Haddad R A. Multiresolution signal decomposition: transforms, subbands, and wavelets[M]. Academic press, 2001.

---

> > ### Author Response · Authors · 2024-12-01
> > **Thanks for follow-up discussion and suggestions**
> >
> > Dear Reviewer uNhX,
> >
> > Since we have provided substantial discussions, comparison and experimental results, we believe the overall quality of our manuscript has been significantly improved according to your valuable comments and suggestions. Now, we are wondering whether our responses and detailed revisions have addressed your concerns. If so, please kindly re-consider your rate of our research. We are looking forward to your support! Thank you.
> >
> > Best,
> >
> > Authors of Paper 705

---

> > > ### Comment · Reviewer_uNhX · 2024-12-02
> > > **Reply to authors' response**
> > >
> > > I appreciate the authors’ further explanation regarding the selection of the auxiliary dataset. However, I still believe this is a crucial aspect of the paper. Without a clear understanding of how the auxiliary data is selected and constructed, it would be difficult for readers to effectively use the proposed model. This lack of insight could limit the applicability of the work, reducing its potential from a general, practical solution to a more narrowly scoped study.
> > >
> > > While the authors argue that time series data can be decomposed into basic components, suggesting that data from different domains with similar decomposed patterns could be helpful, this claim still lacks both theoretical and experimental validation. Specifically, it is unclear what types of decomposed patterns are beneficial, and what degree of similarity would be sufficient for the auxiliary data to contribute effectively. These important points are not addressed in the current version of the paper.
> > >
> > > Furthermore, there is insufficient detail on how to obtain auxiliary datasets with latent similar components. What is the required scale of such datasets, and what criteria should guide their selection? There are still many unanswered questions regarding the auxiliary dataset, which I consider to be a key part of the paper.
> > >
> > > Therefore, I have decided to maintain my current score. I strongly recommend that the authors explore this aspect more thoroughly in future revisions to strengthen the paper and provide more comprehensive guidance for potential users.

---

> ### Author Response · Authors · 2024-12-02
> **Follow-up responses to reviewer uNhX**
>
> Dear Reviewer uNhX,
>
> We would like to further clarify the above two concerns.
>
> - We appreciate your follow-up feedback on the dataset selection. We have clarified that we do not limit the dataset selection when conducting the retrieval process. Actually, we **have provided the selection principle** by exploiting the meta information， which we also consider as **key part of our systematical solution**. Specifically, we filter the auxiliary dataset with **1) same or similar meta information and 2) shapes of time-series**, where the intuition is similar series or series at similar context should provide cues for further forecasting (in A.1.5 of our manuscript). That's to say, our solution has limited the retrieval to same domain/same context and similar temporal periods, which are exactly the **dataset selection principle**.  For example, for the targets of traffic sets, the weather in the same city during similar periods can be an auxiliary dataset where it can be filtered by 'city' and 'time'.  If not available, the similarity-based findings further make sense. Actually, we argue that we **do not impose such compulsory principle** in external dataset construction if such dataset does not exactly exist. If we utilize some principle or criteria to find the required dataset, the dataset satisfying such principle may not exist, it must require quite large-scale efforts to collect, directly leading to the meaningless of such principle. In our opinion, we would like to find the most appropriate dataset in our existing retrieval set, which is also updatable when new datasets arrive. To this end, we believe our solution exactly incorporated such principle as you mentioned (in both **meta information utlization** and **discussion on selection of auxiliary dataset**). Actually, the more similar data, the more beneficial to targets, thus our dataset is updatable.  We also further discuss the potential of pre-clustering strategy on auxiliary set and series assignment in clustering. We believe such strategy can improve both efficiency and effectiveness.
>
> - Regarding **time series data can be decomposed into basic components**, we have provided literature [6,7] to suggest such signal decomposition theory. The fusion mechanism for different components can be diverse. **Actually, the components sharing more similarity with targets are weighted more and the learnable scheme will also force to approach such regularity**, where **the similarity** and **to what extent with similarity** can respectively answer above two questions.  In our work, **what types of decomposed patterns are beneficial, and what degree of similarity would be sufficient** can be learned **with a reinforcement architecture in a data-driven manner**, where the fusion coefficients are discretized and dynamically selected as the fusion hyperparameters should not be fixed. With such data-driven learnable mechanism, the learning algorithm can automatically allow the weighted parameters to approach and align with above principle (more similar, more weighted).
>
>
> To this end, our responses can answer your above two questions, and ZeroTS is scalable and  extendable  to be generalized to  broader application domains with our relaxed and flexible dataset selection principle.
>
>
> Best,
>
> Authors of Paper 705

---

> > ### Comment · Reviewer_uNhX · 2024-12-03
> >
> > Thank you to the authors for their detailed responses. However, while the authors provide some perspective, I still find the arguments insufficiently solid and strong.
> >
> > In the earlier rebuttal, the authors mentioned, "Therefore, we do not specify the relationships between targets and auxiliary bases, and let it select in a natural manner." This time, the principle is stated as "we filter the auxiliary dataset with 1) same or similar meta information and 2) shapes of time-series," but how exactly are "same" or "similar" defined? These relationships need to be **quantitatively measured and experimentally validated**.
> >
> > The statement, "Actually, the components sharing more similarity with targets are weighted more," is still speculative, and there is no experimental evidence to demonstrate that the model operates in this manner. How to quantify similarity and what level of similarity is effective—these questions need to be answered through **empirical analysis**. Additionally, the impact of auxiliary dataset size on experimental results should also be addressed with experiments.
> >
> > To summarize my main concerns regarding the work: First, **the selection and construction of auxiliary datasets need more quantitative analysis and guidance**; second, the use of additional datasets to improve performance will increase **computational resource consumption**, and whether this trade-off is worthwhile in the context of foundation models remains an open question. Therefore, I believe the current version of the work lacks comprehensiveness and applicability.

---

> ### Author Response · Authors · 2024-12-04
> **Follow-up responses to dataset selection concerns**
>
> Dear Reviewer uNhX,
>
> Thanks for final questions. We further clarify and provide some quick  experiments below.
>
> **Formal and quantitive definition of similar/same data records.** The similar meta information can be defined as the same domain, the same city and the limited bias of timestamps ($\Delta t < \tau$), where $\tau$ can be defined as two-hour bias.  The quantitive similarity meaurement between sequences is actually provided in our Eq.(2) in our manuscript.
>
> **Experiments on effectiveness of same/similar meta information selection.**
> We conduct the experiments day and night and report new results on two selected datasets i.e., Traffic and Weather, with  long-term forecasting tasks,  as below.
>
> - Assuming the given  target series is  **Traffic**, we vary the **auxiliary set** as follows and report the MAE for traffic forecasting:
>
>  1)  All set with Traffic: 0.261,   2) All set w/o Traffic:  0.295,  3) All set without weather: 0.267
>
>
> - Assuming the given  target series is  **Weather**, we vary the **auxiliary set** as follows and report the MAE for traffic forecasting:
>
> 1) All set with Traffic:  0.250,  2) All set w/o Traffic: 0.278,   3) All set without weather: 0.257
>
> Actually, we can verify the that  all-set datasets perform best and if the most related domain is removed, the results become decrease.
> And if an unrelated dataset is removed, the performance is near to all-set ones without many influences. We can conclude that the selection strategy is
> reliable and practical.  We further verify the  impact of auxiliary dataset sizes.
>
>
> **The impact of auxiliary dataset size.** Given the target series is Traffic or Weather, we especially vary the scale of datasets in two ways, and report new results on long-term forecasting.
>
> -  **With hierarchical sampling strategy**:  The different categories of auxiliary sets are respectively  reduced by the same proportion.
>
> -  **W/o hierarchical sampling strategy**: Reduce the auxiliary sets  in the random manner by  the proportion.
>
> The performances are as below. We reduce the auxiliary dataset in the order of  20\%, 40\%, 60\%, 80\% of total datasets and report the new results in the following Table:
>
> | MAE                                | Don't reduce | reduce 20% of total datasets | reduce 40% of total datasets | reduce 60% of total datasets | reduce 80% of total datasets |
> |------------------------------------|--------------|------------------------------|------------------------------|------------------------------|------------------------------|
> | With hierarchical sampling/Traffic |        0.261 |                         0.270 |                        0.272 |                        0.281 |                        0.294 |
> | W/o hierarchical sampling/Traffic  |        0.261 |                        0.278 |                        0.281 |                        0.299 |                        0.305 |
> | With hierarchical sampling/Weather |        0.250 |                        0.261 |                        0.275 |                        0.276 |                        0.291 |
> | W/o hierarchical sampling/Weather  |        0.250 |                        0.259 |                        0.274 |                        0.295 |                        0.295 |
>
>
> With our results, we can find that the size can influence the performances especially on without hierarchical sampling but have less impacts with hierarchical sampling. Therefore, there is also the trade-off in dataset scale and performances.
>
>
> Regarding  the  trade-off in the context of foundation models, we believe our ZeroTS is worthwhile against other models, which  has been empirically verified in Table.8 in our revised manuscript. Single PatchTST is  inferior to our ZeroTS and RAG+PatchTST is still inferior to ours.
>
> Actually, as the discussion period is ending, it is unfair for authors to  ask for more empirical study as the final question. Even so, we also thank the reviewer for his follow-up. We are also continuously working day and night on more typical datasets and we will incorporate these more discussions into our revised version if accepted.

---

### Official Review · Reviewer_11xC · 2024-11-02

**Soundness:** 2
**Presentation:** 2
**Contribution:** 2
**Rating:** 5
**Confidence:** 3

**Summary:**

The paper proposes ZeroTS, a zero-shot time series forecasting approach that combines open-world knowledge and data regularities through multi-party data-model interactions. ZeroTS utilizes a TS-RAG (Retrieval-Augmented Generation) module to retrieve meta and series information, allowing domain-specific time series data as prompts. Additionally, the framework introduces a reinforcement learning-based adapter that uses ground-truth feedback to optimize a smaller model iteratively. Extensive experiments show ZeroTS obtains competitive results with reduced memory and inference speed, demonstrating its effectiveness and efficiency in zero-shot and long-horizon forecasting scenarios.

**Strengths:**

1. The paper attempts to bridge the gap between large language models and time series data, which is a relatively unexplored area. This integration could pave the way for new research directions and applications.

2. The proposed method is straightforward and intuitive, and the experiments shown in the paper are comprehensive.

3. The readers can easily understand the proposed method and follow the content of the paper.

**Weaknesses:**

1. One challenge this paper addresses is constructing a time-series-oriented database for RAG construction. However, the authors did not compare their proposed methods with existing fully open-sourced time-series-oriented databases, such as TD-Engine [1]. It is strongly recommended to include this comparison to demonstrate the effectiveness and efficiency of their approach in time-series prompt retrieval.

2. The pre-trained structure of LLMs is heavily reliant on vast amounts of textual training data, which inherently limits their depth of understanding text input. This paper claims to address this limitation by extending LLMs' capabilities to encompass time-series data. However, the input to the LLM remains a sequential representation, making it still difficult for the model to fully capture the input without any further training.

3. Compared to other LLM-frozen baselines, such as Time-LLM, the performance improvements of ZeroTS are incremental and not substantial. A significance test should be conducted to further validate the effectiveness of the model for time series forecasting.

4. In figure3, the paper shows they incorporate Llama2 as the LLM backbone in the paper. However, I did not find any experimental results shown in the paper.

5. The paper overstates its contributions. Constructing a retrieval pipeline for RAG cannot be equated to building a database, as it lacks the fundamental structures, management capabilities, and persistence mechanisms inherent to databases. Properly distinguishing between a retrieval pipeline and a database system is essential to clarify the paper's actual contribution.

[1] https://github.com/taosdata/TDengine

**Questions:**

Please refer to the weakness.

---

> ### Author Response · Authors · 2024-11-20
> **Rebuttal for Reviewer 11xC (1/3)**
>
> Dear Reviewer 11xC,
>
> Thank you very much for providing detailed comments and suggestions to our research. Our detailed responses can be found as below.
>
> **W1. Comparison with TD-engine.** As we know, TD-engine is a industrial-grade database tailored for time-series and dynamic streaming data, to support high-performance, distributed IoT, industrial big data platform [1]. Actually, in our work, as the reviewer said, the core contribution of our work does not lie in the construction of the database system, but the proposed new pipeline for data-driven time-series based retrieval, which couples the emerging RAG technique with LLM-based time-series learning, where the ‘database’ refers to constructing an updatable and collective dataset. Therefore, inspired by your advice, we would like to deploy our solution and algorithm on the open-sourced platform and we also believe the TD-engine can help our solution become more efficient and effective. Thanks for your suggestion, we will modify our expression of ‘database construction’ and further incorporate these discussions in our Related Work, especially the great open-sourced TD-engine system [1] and point out the necessity of achievement transformation in future industrial deployment if this work accepted.
>
> [1] https://github.com/taosdata/TDengine
>
>
> **W2. Clarification on LLM without fine-tuning and superiority of ZeroTS.** We acknowledge the concern raised by reviewer that LLM without fine-tuning leads to less flexibility of language model.
> But actually, fine-tuning the LLM requires substantial sequence observations and large-scale computational costs, this cannot well adapt to the zero-shot time series learning scenarios. To this end, we consider detour the fine-tuning of LLM and propose a lightweight learnable module to remedy the non-fine-tuning of LLM, which allows interactions between real-world external data, LLM model, as well as model error feedback, thus guiding the fusion of retrieved time series and activating the maximal power of LLM. We elaborate the motivation and reasons for our designs.
>
> - *Data availability and computational costs are challenges cannot be neglected.*  LLM is a well pre-trained model with large number of parameters equipped with sufficient semantic patterns within sequences and documents. Therefore, fine-tuning and impacting them will require feeding into large-scale new data and substantial computational costs. However, in in our zero-shot research, the data availability and computational power are big challenges and cannot satisfy enabling the changes of LLM. That’s to say, limited available data can nearly fail to enable changes of parameters in LLM. Let alone for zero-shot learning, it is practically impossible to fine tune LLM (parameters ranging from 700M to 8B) with new arriving data in a real-time manner. Thus, we devise a more lightweight manner which directly utilize the LLM model and learn a small model for prompt construction. Regarding efficiency, we have provided analysis in Table.5 of manuscript and we can observe that our ZeroTS only consumed approximately 1/4 memory overloads and 1/7 inference speed against peer models, indicating its superior efficiency.
>
> - *ZeroTS: Additional inputs as prompts to supplement sequential pattern cues.* To remedy the model adaptation capacity, we especially introduce and preserve an external time-series base (data set), which can be dynamically updated to continuously supplement the new knowledge from real world. To implement this, we couple the emerging Retrieval Augmented Generation (RAG) technique with time-series learning and devise TS-RAG with diverse efficient retrieval strategies. With dynamically updated and retrieved series that share similarity with targeted ones, the series can provide supplementary patterns and leading evolution patterns incorporated into retrieved ones for prompting LLM outputs.
> - *ZeroTS: Active feedback for optimizing targeted and retrieved series representation.* To adapt model with real-world data and ground-truth and equip the model with adaptation capacity, we design a ReinLLM to cohere the feedback from LLM with prepositive series representation learning.
>
> Overall, our ZeroTS is designed tailored for zero-shot time series learning from a new perspective of data-model interactions with a great trade-off, i.e., designing a lightweight learnable ReinLLM adapter. We believe such trade-off has well leveraged the power of LLM and allowed interactions between data and model sides, thus providing sufficient and flexible complementary to existing LLM for series forecasting.

---

> ### Author Response · Authors · 2024-11-20
> **Rebuttal for Reviewer 11xC (2/3)**
>
> **W3. Significance test.**  Compared to other LLM-frozen baselines, such as Time-LLM, our solution is a novel perspective to implement data-model interactions without any fine-tuning on LLM for realizing zero-shot learning. The efficiency comparison against other models can be found in Table.5 in our main text, demonstrating its superiority on both memory and inference speed.
> From the result Table 1 and 2, our ZeroTS can outperform most peer models with extremely low costs. To show the significance, we further conduct the significance tests by comparing ZeroTS against two best baselines Time-LLM and PatchTST on four selected tasks of both zero-shot learning and long-term forecasting. We implement it via t-test with scipy package of Python day and night, and demonstrate the empirical significance of ZeroTS where the p-value is denoted p* when p<0.05.
>
> Performance Table of significance tests for zero-shot learning
>
> |                         |   Time-LLM   |              |   PatchTST   |              |
> |-------------------------|:------------:|:------------:|:------------:|:------------:|
> |                       | p-Value: MAE | p-Value: MSE | p-Value: MAE | p-Value: MSE |
> | ETTh1$\rightarrow$ETTh2 |    0.050*    |     0.065    |    0.045*    |     0.058    |
> | ETTh1$\rightarrow$ETTm2 |    0.048*    |     0.067    |    0.038*    |     0.045*    |
> | ETTh2$\rightarrow$ETTh1  |   0.074    |     0.062    |     0.075    |     0.065    |
> | ETTh2$\rightarrow$ETTh1 |    0.050*    |     0.074    |    0.043*    |    0.040*    |
>
> Performance Table of significance tests for long-term forecasting
>
> |         |    Time-LLM   |          |    PatchTST  |          |
> |---------|:-------------:|:-------------:|:------------:|:-------------:|
> |         | p-Value: MAE  | p-Value: MSE  | p-Value: MAE  | p-Value: MSE  |
> | ETTh1   |    0.0003*    |    0.00025*   |     0.050*    |     0.05*     |
> | ETTh2   |    0.0001*    |    0.0002*    |     0.065     |     0.03*     |
> | Traffic   |    0.075      |     0.063     |     0.050*    |     0.030*    |
> | ILI      |     0.055     |     0.072     |     0.060     |    0.0355*    |
>
> As observed in above two tables, on zero-shot forecasting, the results on 3 tasks out of 4 are of significance, while almost all results are of significance on selected long-term forecasting tasks (9 p-valuer results out of 16 are significant for MSE and MAE) where ZeroTS can exactly outperform Time-LLM and PatchTST on most scenarios by passing t-test. The minor scenarios our ZeroTS is inferior to these two baselines may be the dataset property and the fine-tune process on model adaptation where Transformer layers in our ZeroTS are not fine-tuned.
>
> Thus, we still believe our work is novel insight that enables trade-off among data intelligence, model intelligence and efficiency issue. And our contributions lie in two aspects, i.e., not only the new perspective with RAG for data side adaptation and the ReinLLM coupling reinforcement learning with LLM to facilitate the optimization of a small representation model, but also the ultra-lightweight learnable parameters in our ZeroTS, providing possibility in edge computing and resource-limited services.

---

> > ### Author Response · Authors · 2024-11-20
> > **Rebuttal for Reviewer 11xC (3/3)**
> >
> > **W4. Exploitation of Llama2 version LLM.** We are sorry for such confusion. The Llama2 in Fig.3 should be corrected as GPT-2. All results of ZeroTS reported in our manuscript are implemented with GPT-2. Actually, in our pre-experiments, we have ever conducted experiments with both GPT-2 and Llama2, and unfortunately, we find that Llama2 is equipped with large-scale parameters that hinder the inference speed of ZeroTS. But the performance difference between Llama2 and GPT-2 is not significant. Considering a resource-friendly and efficient solution, we finally choose GPT-2 as the backbone and we have corrected the label of Llama2 in Fig.3 into GPT-2. Thanks for your reminding of this typo!
> >
> >
> > **W5. Overstatement of contributions.** Thanks for pointing out this misunderstanding! Actually, we have not emphasized the contribution of building a time-series database in the Contribution Section (in Introduction part) of our manuscript. Conversely, the same with the reviewer, we acknowledge and agree that constructing a retrieval pipeline for RAG cannot be equated to building a database. We also consider that constructing a new database is not the contribution belong to the ranges of ICLR community. Instead, in this research, we establish a new pipeline for time-series based retrieval and devise a series of techniques to couple the RAG with time-series zero-shot learning. In Contributions of Introduction, our claim can be found as ‘We first introduce RAG into zero-shot time-series prediction, devise a meta-value structure to accommodate informativeness of various series, and a modified quick retrieval HSNSW with hybrid affinity measurement for efficient retrieval.’ At this time, we speculate that the expression of ‘database’ may mislead the understanding of our technical contribution for reviewer. We will instead ‘database’ into ‘collective data set’ or ‘an auxiliary set of series from diverse domains’ in our revised manuscript in both ‘Challenge’ part and ‘Method’ part to avoid confusion and misunderstanding. Thanks for this point and we have revised the manuscript.
> >
> > Best,
> >
> > Authors of Paper 705

---

> > > ### Author Response · Authors · 2024-11-23
> > > **Further discussion**
> > >
> > > Dear Reviewer 11xC,
> > >
> > > Thank you very much for devoting time on reviewing our manuscript. As the author-reviewer discussion process is ending soon, we are wondering whether our responses have well addressed your concerns. If you have any further questions regarding our manuscript, please let us know and we are glad here to provide further discussion and clarification to improve the quality of this manuscript. Thanks.
> > >
> > > Authors of Paper 705

---

> > > > ### Author Response · Authors · 2024-11-28
> > > > **A brief rebuttal for Reviewer 11xC**
> > > >
> > > > Dear Reviewer 11xC,
> > > >
> > > > Thank you very much for providing detailed comments and suggestions to our research. We have summarized the responses and changes in brief for your reviewing, and you can also refer to full version to check the details.
> > > >
> > > >
> > > > **1) Comparison with TD-engine.** TD-engine is an **industrial-grade database** tailored for time-series and dynamic streaming data, to support high-performance, distributed IoT, industrial big data platform.  The **core contribution** of our work is the proposed **new pipeline for data-driven time-series based retrieval, which couples the emerging RAG with LLM-based time-series learning**, where the ‘database’ refers to constructing **an updatable and collective dataset**.  To this end, we further provide discussions on practical deployment of ZeroTS with TD-engine in the Future work subsection.
> > > >
> > > > **2) Why not fine-tune LLM:** Data availability and computational costs are challenges cannot be neglected.  And we believe our ZeroTS enjoys the following two advances.
> > > >
> > > > **Superiority of ZeroTS:**  1) Additional inputs as prompts to **supplement sequential pattern cues**.  2)  **Active feedback** for optimizing targeted and retrieved series representation.
> > > >
> > > > **3) Significance test.** We have provided the detailed significance tests against best baselines. We have further provided discussions on our significant tests, new ways of data model intelligence and their interactions towards time-series forecasting.
> > > >
> > > > **4) Exploitation of Llama2 version LLM.**  The Llama2 in Fig.3 should be corrected as GPT-2.
> > > >
> > > > **5) Overstatement of contributions.** Actually, we **agree** that constructing a retrieval pipeline for RAG cannot be equated to building a database  and constructing a new database **is not the contribution belong to  ICLR community**. Instead, in this research, we establish a new pipeline for time-series based retrieval and devise a series of techniques to couple the RAG with time-series zero-shot learning. We **have replaced ‘database’ with ‘collective data set’ or ‘an auxiliary set of series from diverse domains’ in our revised manuscript** in both ‘Challenge’ part and ‘Method’ part to avoid such misunderstanding.
> > > >
> > > >
> > > > Thank you very much for devoting time on reviewing our manuscript. We have revised our manuscript and carefully incorporated these additional discussions into our manuscript. We wonder whether our revisions and responses have well addressed your concerns. If you have any further questions, we are happy to provide more discussions and explanations. Thank you.
> > > >
> > > > Best,
> > > >
> > > > Authors of Paper 705

---

> > > > > ### Author Response · Authors · 2024-12-03
> > > > > **Further discussion inquires**
> > > > >
> > > > > Dear Reviewer 11xC,
> > > > >
> > > > > Thank you very much for devoting time on reviewing our manuscript. As the author-reviewer discussion process is ending soon, we are wondering whether our responses have addressed your concerns. Please kindly provide the feedback to our responses and if you have further concerns, we are also glad to further discuss these issues.  We sincerely hope you can kindly support this research and thank you very much!
> > > > >
> > > > > Best regards,
> > > > >
> > > > > Authors of Paper 705

---

### Official Review · Reviewer_eiSN · 2024-11-03

**Soundness:** 3
**Presentation:** 4
**Contribution:** 3
**Rating:** 6
**Confidence:** 4

**Summary:**

The paper presents ZeroTS, a novel zero-shot time series prediction model designed to operate without task-specific fine-tuning, addressing data scarcity and adaptation across multiple domains. It introduces a unique framework for zero-shot time-series forecasting by leveraging Retrieval-Augmented Generation (RAG) to retrieve relevant time-series data and combining this with reinforcement learning to optimize interaction between a smaller learnable module and a large language model (LLM). Key contributions include a hybrid retrieval framework to address similarities in time series data and an efficient ReinLLM adapter, which integrates error feedback for more accurate predictions. Experiments show that ZeroTS provides competitive zero-shot and long-horizon forecasting performance while remaining parameter-efficient and memory-friendly.

**Strengths:**

- The paper presents a novel combination of RAG and reinforcement learning for zero-shot time series forecasting, which is an interesting and promising direction.
- ZeroTS demonstrates strong empirical performance on various benchmark datasets, achieving state-of-the-art results in zero-shot forecasting and competitive results in long-horizon forecasting.
- ZeroTS exhibits superior efficiency compared to existing LLM-based methods, requiring significantly less memory and achieving faster inference speeds, making it more practical for real-world applications.
- The ablation study substantiates the contributions of key components such as RAG, the ReinLLM adapter, and learnable parameters, strengthening the overall argument for the system’s design choices.
- The case study provides an intuitive understanding of how retrieved series can help improve prediction accuracy and mitigate LLM hallucinations.

**Weaknesses:**

- The use of GPT-2 as the LLM backbone might be considered outdated, given the rapid advancements in LLM technology. Using a more recent and powerful LLM could potentially further enhance performance.
- While the authors address the complexity of the retrieval process, it remains a potential bottleneck for the scalability of the system. Further investigation into more efficient retrieval strategies is warranted.
- The paper acknowledges that ZeroTS shows weaker generalization when transferring from longer to shorter forecasting horizons. This aspect needs further investigation and improvement.
- The reliance on extensive time-series data for retrieval raises concerns about scenarios where domain-specific data is scarce or unavailable, potentially impacting generalizability.
- The algorithm description in Algorithm 1 could benefit from additional clarity and elaboration. Providing more specific details on the interaction between the policy and value networks within the ReinLLM adapter would enhance understanding.
- While a hyperparameter study is presented, a more comprehensive discussion of the impact of different hyperparameters on performance across various datasets would strengthen the paper.

**Questions:**

- How would the performance of ZeroTS be affected by using a more recent and powerful LLM, such as GPT-3 or Llama-3?
- What are the limitations of the proposed ρ-ρ-cut-off strategy for reducing retrieval complexity? Are there alternative strategies that could be explored for further efficiency improvement?
- What are the specific reasons behind the weaker generalization observed when transferring from longer to shorter prediction horizons? Are there potential modifications to the model architecture or training procedure that could address this issue?
- Could you provide a more detailed explanation of Algorithm 1, specifically focusing on the interaction between the policy network, the value network, and the update process of the series representations within the ReinLLM adapter?
- Could you elaborate on the choice of hyperparameters and their impact on performance across different datasets? Were there any specific challenges encountered during hyperparameter tuning?

---

> ### Author Response · Authors · 2024-11-20
> **Rebuttal for Reviewer eiSN (1/3)**
>
> Dear Reviewer eiSN,
>
> Thanks for providing a high rating on our manuscript and insightful question for improving the quality of our manuscript. We have provided more details, experiments and discussions to dispel your concerns and tried our best efforts to satisfy the high standards of ICLR community. The responses can be found as below.
>
> **W1&Q1. More recent alternative LLM for testing**. Actually, we would like to clarify that why we chose GPT-2 as the backbone. First, to avoid the data leakage issue and fairly validate the effectiveness of our ZeroTS, we choose GPT-2 (released in 2019) where the training datasets including traffics, ETTh1-ETTm2, and other datasets are all released after 2019. Second, GPT-2 is a relatively lightweight LLM with 700M parameters, while other LLM are in billion levels. The experimental results have exactly demonstrated the rationality of our intuition and whole learning process. It is fine to replace GPT-2 with any other LLM, such as Llama-3. We now have conducted experiments day and night, and report the new results of MSE by combining RAG and Llama-3 (8B Para. for inference), also combining RAG with a conventional foundational model PatchTST for verifying the necessity of LLM, as following.
>
> |       MSE     | RAG+Llama-3 | RAG+PatchTST | RAG+GPT2 | PatchTST |
> |:-----------------------:|:-----------:|:------------:|:--------:|:--------:|
> | ETTh1$\rightarrow$ETTh2 |    0.345    |    0.361    |   0.350  |   0.380   |
> | ETTh1$\rightarrow$ETTm2 |    0.274   |    0.286    |   0.272  |   0.314  |
> |          ETTh1        |    0.402    |    0.410   |   0.403  |   0.413  |
> |          ETTh2        |    0.312    |    0.319    |   0.315  |   0.330   |
>
> It is noted that a more powerful LLM can improve the performances maximumly by 1.42\% (obtained on ETTh1$\rightarrow$ETTh2 scenario from 0.350 to 0.345, others are ranging from 0.2\% to 0.9\%), but the inference efficiency of larger LM (e.g., Llama-3: 8 Billion) has dropped significantly. When replacing LLM with PatchTST, the performance will drop by 1\% to 5\% (maximumly arrive at ETTh1$\rightarrow$ETTm2 from 0.272 to 0.286). We can witness the improvement of LLM is relatively deserve for its efficiency and enhance the choice of GPT-2 in ZeroTS.
>
> Thus, we can conclude that our GPT-2 is sufficient for our task and more powerful LLM can further sacrifice efficiency which may not deserving for our predictions.
>
> **W2&Q2. Limitations of the proposed $\rho$-$\rho$-cut-off strategy and Alternative strategies for improvement of retrieval.** $\rho$-$\rho$-cut-off is a trade-off between the global retrieval and local retrieval, which has already considered the efficiency issue from $M$ to $\rho^2 \times M$, where $\rho$ controls the complexity. A potential alternative way towards the retrieval process can be pre-clustering the series prototypes, namely prototype-based retrieval. When the auxiliary series base constructed, we can first pre-cluster all the series into different clustering where one prototype can represent a corresponding clustering. When a new series is added into the base, it will be assigned into one clustering and corresponding prototype can be updated. To this end, given a targeted series, we can retrieve auxiliary series by matching target one to prototypes and reduce the complexity from $\rho^2 \times M$ to linear to M$\times$log(N), where $N$ is the number of total series in auxiliary base, $M$ is the number of clustering of total series.
>
> **W3&Q3. Overcoming the transfer from longer horizons to shorter ones.** We speculate the specific reason behind this phenomenon is the changes of forecasting granularity, i.e., when the prediction resolution is improved, even though backbone patterns are shared, the model cannot immediately adapt previous coarse-grained evolutions to fine-grained ones, where the fine-grained evolution is with more details of element-wise temporal dependencies while coarse-grained ones are missing. Namely, when up-sampling the pixel-level sequence resolution, the details will be missing thus leading to relatively reduced performance.
>
> On overcoming such challenge, we provide some discussions on two aspects. First, we can design **a multi-resolution prediction** in training period, and enable the model to aware the potential multi-grained evolution patterns and potential interpolation in the sequential elements with new learning objective, i.e., outputs on different temporal granularities. Then when the transfer is implemented, the coarse-to-fine pattern can be potentially transferred from known granularity to unknown ones. Second, we can devise **temporal interpolation strategy and sequence augmentation** to imitate series in fine-grained resolution, e.g., exploring the linear interpolation or spline interpolation for filling sequential elements thus contributing to the adaptation on transferred domain. Thanks for your insightful question and we will incorporate these discussions in our Appendix!

---

> ### Author Response · Authors · 2024-11-20
> **Rebuttal for Reviewer eiSN (2/3)**
>
> **W4. The reliance on extensive time-series data.** Actually, we utilize the external time-series data for retrieval but do not specify any domain. As illustrated in Fig.1 in our Introduction, we exploit the similarity of series shape and evolution trends as the cues for retrieval. The meta information including domain category is also taken as an element for retrieval but not compulsory. Therefore, our retrieval will not raise concerns on data scarcity issue and will not impact the generalization ability. Thanks for this question, and we will emphasize our retrieval strategy that does not specify on domains.
>
> **W5&Q4. More detailed explanation of Alg.1.**
> Thanks for your kind remind! The Alg. 1 is associated with the framework architecture (the flow direction of data) in Fig. 3 of main text. Specifically, given a targeted time-series, we activate the retrieval process with an efficient heuristic ($\rho-\rho$-cut-off) and hierarchical (Hierarchical Series-level Navigable Small World, HSNSW) strategy to obtain a series of auxiliary series for prompt construction. The **policy network** receives target series and auxiliary series as input, and takes the role of selecting the critical hyperparameters for targeted time-series representation and auxiliary time series fusion. **Update process.** When the action (important hyperparameter) selected, the representation will be updated with new hyperparameters, and then feed the updated representation into LLM, where targeted representation as main input to LLM and fused auxiliary series as prompt to LLM for supplementary cues. After that, TS-LLM, instantiated by GPT-2, outputs the prediction results and deliver the outcome and errors to the value network. Our **value network** transfers the errors into reward and backpropagate to the policy network for strategy optimization. Actually, our lightweight learnable module is optimized from both data side and model side. On data side, we exploit retrieval process from auxiliary real-world data, and let updatable real data to continuously supplement the cues for model prediction. And on model side, the model (LLM) is considered as the environment and we let our lightweight learnable module ReinLLM interact with LLM and derive the error feedback as reward to optimize the whole process, contributing to the real-world corrections and enabling seamless data-model-world interactions.
>
> We will involve these explanations linked with Alg.1 in our Appendix. Thanks!

---

> ### Author Response · Authors · 2024-11-20
> **Rebuttal for Reviewer eiSN (3/3)**
>
> **W6&Q5. Choice and impacts of hyperparameters on our model.**
>
> The vital hyperparameters of our model can be five-fold illustrated in Table.6, number of retrieved auxiliary series $K$, representation dimensions of both target series and auxiliary series, cut-off coefficient $\rho$, step size for action selection $\eta$, as well as dimension of retrieved series. We consider that the specific **challenge** for hyperparameter tuning lies in the complexity of parameter combination, leading to combination explosion. To address this challenge, we integrate the intuitive analysis with small data pre-exploration and progressively approaching strategy to alleviate the searching burden. For example, we first find that Top-$K$ series play a more significant role in our model, then we test different $K$ on a relatively small scale dataset such as ETTh1, and then find the potential optimized ranges of $K =\{3,6,9,12\}$. And then we can fix the $K$ to optimize other hyperparameters. When one parameter achieves relatively optimal, we can exploit such parameter for further tuning of others. Finally, we report our fine-tuning process in Fig.4 and Fig.6 in our manuscript and present the final results in Table. 6. We hope these explanations can help you better understand the process. We elaborately discuss them as below.
>
> -	The most important one is the number of retrieved auxiliary time series, which is concerned with both effectiveness and efficiency. The more series are retrieved, the more informativeness of prompts in LLM and increases the overall performance and generalization. But with more information retrieval, the computational costs on retrieval and fusion become higher. We can see that when $K$ arrives 6, our ZeroTS becomes pretty good performances and satisfies the trade-off.
> -	Regarding representation dimensions for both targeted and retrieved series, we let them range from ${8, 16, 24, 32, 64 }$ where it also requires the trade-off. It is known that larger dimensions usually account for more learning and fitting capacity, which corresponds to factual performance illustrated in Fig. 4. We thus take 64 as the final dimension as larger dimension will significantly increase the computational burden, e.g. 128.
> -	The coefficient for cut-off retrieval $\rho$. Such parameter determines which ranges of retrieved series can be obtained by our model and input into LLM as prompts. The larger $\rho$, suggests more available series, and thus indicate more cues and evolution patterns obtained for LLM, yielding increasing model performances. To achieve the trade-off, we search the optimal one ranging from $\{0.2, 0.3, 0.4, 0.5\}$ and find 0.3 or 0.4 as optimal point for respective datasets. Actually, finer granularity can be explored but it requires more costs.
> -	Step size for action selection $\eta$. It is the parameter for discretizing the coefficient $\alpha$ selection process in Eq.(7) thus for its adjustment. Since the coefficient $\alpha$ ranges from 0 to 1, the step stride is selected from $\{0.05, 0.1, 0.15, 0.2\}$ where larger step can lead to leap over optimal point while smaller step leads to inefficient dilemma. Finally, according experimental analysis and tests, we choose $\eta = 0.05$.
>
> Thank you for recommending above discussion to improve the understanding of our ZeroTS.
>
>
> Finally, we would like to thank you again for your insightful suggestions to supplement more discussions regarding model selection, retrieval process, potential further strategies and hyperparameter settings, thus significantly improving the overall quality of our manuscript.
>
> Best,
>
> Authors of Paper 705

---

> > ### Author Response · Authors · 2024-11-23
> > **Further discussion**
> >
> > Dear Reviewer eiSN,
> >
> > Thank you very much for devoting time on reviewing our manuscript. As the author-reviewer discussion process is ending soon, we are wondering whether our responses have well addressed your concerns. If you have any further questions regarding our manuscript, please let us know and we are glad here to provide further discussion and clarification to improve the quality of  this manuscript. Thanks.
> >
> > Authors of Paper 705

---

> > > ### Author Response · Authors · 2024-11-28
> > > **A brief rebuttal for Reviewer eiSN**
> > >
> > > Dear Reviewer eiSN,
> > >
> > > Thanks for providing a high rating on our manuscript and insightful question for improving the quality of our manuscript. We have highlighted the responses and changes in brief for your reviewing, and you can also refer to full version to check the details.
> > >
> > > **1) Why choose GPT-2 and how about recent alternative LLM.**
> > >
> > > - **Why GPT-2?** 1) Avoiding information leakage issue on testing. 2) With lightweight parameters (GPT-2: 700M Para. vs Llama-3: 8B Para. for inference).
> > >
> > > - **Llama-3 testing.** We have conducted the additional experiments on Llama-3 and a few improvements have been demonstrated. Details can be found in our detailed responses and revised version
> > >
> > > **2) Limitations of proposed $\rho-\rho$ cut strategy and Alternative strategies for improvement of retrieval.** The shortcoming may lie in the sampling controlled by $\rho$, introducing the randomness. An alternative can be pre-clustering all auxiliary series as prototypes and match the prototype for following index.
> > >
> > >
> > > **3) Overcoming the transfer from longer horizons to shorter ones.** The reason may lie in is the changes of forecasting granularity.  **Two potential improvements.** 1) Multi-resolution prediction for forecasting. 2) Temporal interpolation and sequence augmentation to imitate series in fine-grained resolution.
> > >
> > >
> > > **4) The reliance on extensive time-series data.**  We do not require intensive target domain related series for retrieval. The intuition is that time-series with similar or related patterns can be reinforced for forecasting with each other and series can be decomposed into different components. We can find the Top-K series and dynamically combine them in a learnable manner, which further provides the cues of potential trends for output of the backbone language model.
> > >
> > > **5) More detailed explanation of Alg.1.** As shown in Fig.2, it undergoes a hierarchical retrieval process and then we impose the learnable representation learning and fusion scheme in our ReinLLM. With representations of targeted and retrieval series ready, we can input these representations with retrieved series as prompts into LLM, and exploit the errors between LLM and ground-truth to backpropagate the adjustment to previous neural network. Details can be found in our revised manuscript and Fig.2.
> > >
> > > **In detail**, **policy network** receives target series and auxiliary series as input, and takes the role of selecting the critical hyperparameters for targeted time-series representation and auxiliary time series fusion. **Update process.** When important hyperparameter selected, the representation will be updated with new hyperparameters, and then feed the updated representation into LLM, where targeted representation as main input to LLM and fused auxiliary series as prompt to LLM for supplementary cues. After that, TS-LLM, instantiated by GPT-2, outputs the prediction results and deliver the outcome and errors to the value network. **Value network** transfers the errors into reward and backpropagate to the policy network for strategy optimization.
> > >
> > >
> > >  **6) Choice and impacts of hyperparameters on our model.** The specific challenge for hyperparameter tuning may be the complexity of parameter combination, leading to combination explosion. We can take **a small batch of data** for pre-exploration and progressively approach the optimal to alleviate the searching burden. The fine-tuning of hyperparameters is usually the **trade-off between efficiency and effectiveness**. We have detailed such trade-off in our experiments of manuscript and responses.
> > >
> > > Thank you for recommending above discussion to improve the understanding of our ZeroTS. We **have carefully revised our manuscript and incorporated these additional discussions into our manuscript**. If you have further questions, please feel free to engage the discussion and we are happy to further improve our manuscript. Thank you.
> > >
> > > Best,
> > >
> > > Authors of Paper 705

---

> ### Author Response · Authors · 2024-11-30
> **Reporting some new exciting results following your insightful suggestions**
>
> Dear Reviewer eiSN,
>
> Thanks for your kind support on our research. As the discussion period is ending soon, we are excited to **report some new experimental results following your insightful suggestions**.
>
> We have conducted some **new experiments by hierarchical prototype-based retrieval**. We first cluster all series into different clusterings, e.g., M=6 in our auxiliary datasets, where the **clustering centers can be viewed as  the prototypes** for retrieval. The detailed retrieval on individuals is followed-up.  We then select the most and second matching prototypes and then retrieve two series at each selected clustering. Then four series are incorporated as our  prompts feeding into TS-LLM,  **allowing more series to be retrieval without sacrificing too much computations**.
>
> The **performances on zero-shot prediction (MAE)** and **speed of inference process (s/iter)** can be found as below,
>
> - **Performances on zero-shot prediction  with hierarchical retrieval** on four scenarios:
>
> ETTh1→ETTh2  0.380, ETTh1→ETTm2 0.325, ETTh2→ETTh1  0.520,  ETTh2→ETTm2  0.328
>
> - **Inference Speed  with hierarchical retrieval** on ETTh1-96/ETTh1-336: ETTh1-96 :    0.032 (s/iter), ETTh1-336:   0.054 (s/iter)
>
>
> As observed, the performances **do not show much difference from our ZeroTS** while the **efficiency is significantly improved**.
>
> Since the clustering is **once for all** and when new series arrive, it can be dynamically inserted into either one clustering. When the patterns change a lot, the clustering can be updated. And it also satisfies the **updatablity and flexiblity in our data-model  interaction (the core claim of ZeroTS)**.
>
> For **cross-granularity zero-shot transfer**,  we are going to leave it as our **future work**, which will incorporate more advancing technologies such as context-aware prompts, data adaptive sampling and so on for more improvements. Thanks very much for inspiring us into further thoughts.
>
> Thanks again for supporting our research and promoting the overall quality of the manuscript! We will incorporate these discussions into our further version!
>
>
> Best,
>
> Authors of Paper 705

---

### Official Review · Reviewer_fTsg · 2024-11-04

**Soundness:** 3
**Presentation:** 3
**Contribution:** 3
**Rating:** 8
**Confidence:** 3

**Summary:**

This paper presents ZeroTS, a framework for zero-shot time-series prediction that integrates retrieval-augmented generation (RAG) with time-series data, addressing challenges in this area. The authors highlight the difficulties in adapting RAG for time series, including the need for an effective time-series-oriented database, optimization of the dual pipeline for high-quality outputs, and the construction of a lightweight module to integrate knowledge from both time-series data and large language models (LLMs):

1. The authors utilize RAG for the time series prediction. ZeroTS is claimed to be the first framework to apply RAG to zero-shot time-series prediction, incorporating a meta-value structure to capture key information across time-series data. A modified high-speed nearest neighbor search (HSNSW) with hybrid affinity measurement enables efficient and targeted retrieval, enhancing prediction relevance. The authors also propose a distant metric that combines both the cosine similarity and Euclidean distance.

2. The authors propose a reinforcement scheme that refines the coherence between retrieved data and LLM output. This scheme leverages feedback from the ground-truth data to guide the LLM's predictions closer to factual trends, enabling the model to learn and adjust to real-world patterns dynamically.

3. ZeroTS is evaluated in both zero-shot and long-term prediction settings, demonstrating performance gains while outperforming comparable models with better space allocation and computational time.

**Strengths:**

1. The paper is written clearly. The organization of the paper is well understood and the authors illustrate their algorithms with decent figures.

2. The authors utilize RAG for the time series prediction. Their way of constructing the retrieval database seems reasonable. In addition to RAG framework, the authors propose a reinforcement learning-based algorithm to use the retrieval dataset in an efficient way.  This combination enables the authors to make time series prediction in zero-shot.

3. The authors conduct a reasonable amount of ablation studies to observe the effect of their algorithms. Additionally, the authors study the effect of their selection of hyperparameters: the number of retrieved time series, the representation dimension for the retrieval and target domains.

**Weaknesses:**

1. I think the reinforcement learning-based algorithm the authors propose has a similarity with the literature of reinforcement learning with human feedback (RLHF)[1] [2] in the sense that this proposed algorithm is trying to learn the policy based on the retrieved time series (feedback). It would be better to see the similarity and differences between the proposed algorithm and RLHF literature. I am aware that there is a page limit but this discussion can be performed in Appendix as well.

2. The authors present their results by averaging over prediction horizons. It would be more informative to include results for each prediction horizon in the appendix, allowing readers to assess model performance across specific horizons. This approach would also address any concerns about one horizon dominating and skewing the overall average.

3. Minor Weakness: In line 232, the authors claim that they adopt a hybrid distance metric by incorporating both cosine similarity for trend modeling and Euclidean distances for value discrepancy. I think the definition in (2) does not provide a distance notion, instead it provides a similarity notion. This is because this definition is positively correlated by the cosine similarity and negatively correlated by the Euclidean distance.

4. The authors keep saying that they utilize meta information. However, I think, the way of utilizing meta information in the paper is not clear. I asked questions about this weakness in the questions section.


[1] Ouyang et al. Training language models to follow instructions with human feedback

[2] Rafailov et al. Direct Preference Optimization: Your Language Model is Secretly a Reward Model

**Questions:**

1. How do the authors utilize meta information during retrieval phase? In line 214-215, the authors write that "We select the
domain category, timestamps and spatial location or user identification as the meta information, and calculated mean and variance are also considered meta information for similarity-based retrieval.". However, I could not find any point in the retrieval that the authors utilize meta information other than mean and std statistics. Could the authors point out where they used domain category, timestamps and spatial location or user identification as meta information during the retrieval phase?

2. In figure 2, the authors illustrate their algorithms and there is open-world textualizing knowledge in the top-right of the figure. Similar to the previous question, I could not understand where the authors specifically utilize meta information in ReinLLM Adapter phase. In line 303, the authors write that "In detail, the meta information is tokenized and concatenated by textual attributes into $\tilde{X}_m$.". Specifically, which meta information do the authors concatenate? How do the authors obtain these meta information and tokenize them?

---

> ### Author Response · Authors · 2024-11-20
> **Rebuttal for Reviewer fTsg  (1/2)**
>
> Dear Reviewer fTsg,
>
> Thanks very much for your positive feedback, we have provided additional discussions and responses to your concerns.
>
> **W1. Differences and similarity between ReinLLM in ZeroTS and RLHF.**　Actually, both ZeroTS and RLHF utilize reinforcement scheme to enhance the model performance. However, they are different on which side of information to receive and how to implement further optimization.
> Specifically, ZeroTS receives the feedback from LLM and optimizes the representation of fused retrieved time series, where the feedback can be automatically computed via the differences between ground-truth and LLM output, and it enjoys the nice property of delivering the model side feedback to previous data side. In contrast, RLHF in InstructGPT explores a fine-tune process (PPO) to align the awareness between GPT and human annotated labels [1]. DPO [2] exploits a scheme to remove the unstable reinforcement process and direct optimizes the reward from human label rewards.
> To this end, RLHF requires intensive data and extensive human labors for LLM fine-tune, while our ZeroTS optimizes the retrieved series (both representation and fusion mechanisms) fusion with gaining awareness from LLM output side, which is more efficient, automatic, and lightweight with fewer human workloads, yielding more practicality.
>
>
> **W2. Results by prediction horizons.** Thanks for remind. Drawing the results for each prediction horizon brings more informativeness and helps better understanding of our model. Actually, we utilize the retrieved time series with longer horizons as prompts to LLM, thus it can provide longer sights for prediction, avoiding one horizon dominating and skewing. To dispel your concern, we report the results for each horizon on long-term forecasting and zero-shot forecasting, where results of four setting are reported in respective tasks. ETTh1, ETTh2, ETTm1, ETTm2 for long-term prediction and ETTh1$\rightarrow$ETTh2, ETTh1$\rightarrow$ETTm2, ETTh2$\rightarrow$ETTh1, ETTh2$\rightarrow$ETTh1 for zero-shot forecasting. We report the horizon-level MAE of ZeroTS on common setting, in the order of 96/192/336/720/Average. The full comparison can be found in our revised manuscript.
>
> For zero-shot prediction, the results of MAE can be found as below,
>
> ETTh1$\rightarrow$ETTh2: 0.336/0.368/0.396/0.417/0.379
>
> ETTh1$\rightarrow$ETTm2: 0.271/0.310/0.341/0.387/0.327
>
> ETTh2$\rightarrow$ETTh1: 0.485/0.501/0.544/0.558/0.522
>
> ETTh2$\rightarrow$ETTh1: 0.261/0.324/0.365/0.391/0.335
>
> For long-term prediction, the results  of MAE can be found as below,
>
> ETTh1: 0.381/0.399/0.438/0.466/0.421;   ETTh2: 0.325/0.373/0.391/0.410/0.375
>
> ETTm1: 0.345/0.359/0.384/0.421/0.377;  ETTm2: 0.257/0.320/0.361/0.387/0.331
>
> As observed in comparison with others reported in Ref.[3], we can conclude our solution outperforms baselines on most horizons, and **there is no skewing on one horizon**, which can dispel such concern. We will incorporate these results into Appendix of our revised manuscript. Thanks again for your kind remind.
>
> [3] TIME-LLM: TIME SERIES FORECASTING BY REPROGRAMMING LARGE LANGUAGE MODELS, Ming Jin, et, al. ICLR 2024.
>
> **W3. Distance notation.** Thanks for your remind! Eq.(2) is exactly a similarity measurement, and it can be viewed as the inverse of distance, the larger, the proximity between two series are higher and thus the distance is closer. We will revise the distance to similarity, and denoted as ‘${\rm Sim}(\cdot, \cdot)$’.

---

> ### Author Response · Authors · 2024-11-20
> **Rebuttal for Reviewer fTsg (2/2)**
>
> **W4&Q1. Meta information utilization.** The meta information table can be found in Table.4 of our Appendix. Regarding the retrieval process, we utilize the meta information including domain category (e.g., traffics/electricity/weather…), timestamps (2024/11/11/10:00) and spatial location/user identification to help retrieve related auxiliary series. Of course, calculated mean and variance are also considered meta information for similarity-based retrieval. Specifically, given a targeted series with its domain category, timestamps, and identification (user/location id.) where it is available when data is collected, we can retrieve time series from auxiliary series base by indexing corresponding meta information (keywords), if the meta information is matched, e.g., two series share the same domain category, share similar timestamps (day of week, hour of day), this series will be selected. And we can compute the similarity between selected series and targeted series, where the top-K similar series will be utilized for following fusion and representation as prompts.
>
> **W4&Q2. Meta information utilization in ReinLLM.** Regarding our ReinLLM, meta information is considered as the features of retrieved series, which consists of domain category, timestamps and data description, and we take the concatenation of meta information, trend vector as well as deterministic series values as the integrated representations. The abovementioned meta information (domain category, timestamps and data description) is available when the auxiliary base is constructed, and it corresponds to the dataset and series-level property. Then the words are transferred into vectors with the tokenizer in LLM (i.e., GPT-2 in our ZeroTS), where the tokenizer is trained by the strategy named Byte-Pair Encoding (BPE), i.e., the tokenizer trained with occurrence frequency of words and their semantics.
>
> Thanks for pointing out these issues and we will add more details into our revised manuscript to make our research as clear as possible.
>
> Best,
>
> Authors of Paper 705

---

> ### Author Response · Authors · 2024-11-23
> **Further discussion**
>
> Dear Reviewer fTsg,
>
> Thank you very much for devoting time on reviewing our manuscript. As the author-reviewer discussion process is ending soon, we are wondering whether our responses have well addressed your concerns. If you have any further questions regarding our manuscript, please let us know and we are glad here to provide further discussion and clarification to improve the quality of  this manuscript. Thanks.
>
> Authors of Paper 705

---

> ### Author Response · Authors · 2024-11-28
> **A brief rebuttal for Reviewer fTsg**
>
> Dear Reviewer fTsg,
>
> Thank you very much for your positive feedback as well as great efforts on reviewing our manuscript. We have summarized the responses and changes in a short version as below for your reviewing.
>
> **1) Differences and similarity between ReinLLM in ZeroTS and RLHF.** Both ZeroTS and RLHF utilize reinforcement scheme to enhance the model performance, but they are different in which side of information to receive and how to implement further optimization.
>
> -	ZeroTS receives the feedback from LLM while RLHF receives feedback from human.
> -	ZeroTS adjusts the learnable module with ground-truth and does not fine-tune the LLM, while RLHF requires intensive data and extensive human labors for LLM fine-tune.
>
> **2) Horizon-level predictions.** We **have reported** the horizon-level results on both zero-shot and long-term settings in our responses  and revised manuscript. We can conclude our solution outperforms baselines on most horizons, and **there is no skewing on one horizon**, which can dispel such concern.
>
> **3) Distance notation.** We have revised the ‘distance’ into ‘similarity’.
>
>
> **4) Meta information utilization.**  In **retrieval process**, we utilize meta information for domain-specific matching, and filtering the series with similar context (nearest hour of day, day of week and so on) for fusion.  In **ReinLLM**, we take meta information as features for representation and we transfer meta elements into vectors with the tokenizer in LLM (Each LLM has its own tokenizer,  Byte-Pair Encoding (BPE) in GPT-2).
>
> Thanks for pointing out these unclear issues and **we have added these above details into our revised manuscript**. We also wonder whether our responses and revisions have addressed your concerns. If you have any further questions, we are always glad to provide more discussions and explanations. Thank you.
>
>
> Best,
>
> Authors of Paper 705

---

> > ### Comment · Reviewer_fTsg · 2024-11-29
> >
> > I want to thank the authors for the detailed reply. Since my concerns have been addressed, I will maintain my positive score.

---

> > > ### Author Response · Authors · 2024-11-29
> > > **Thanks for your kind follow-up**
> > >
> > > Dear Reviewer fTsg,
> > >
> > > Thanks for your great help on improving our manuscript and kind follow-up. We will continue to refine and polish our manuscript to satisfy the high quality of ICLR community. Thanks again!
> > >
> > >
> > > Best,
> > >
> > > Authors of Paper 705

---

### Author Response · Authors · 2024-11-21
**Summary of changes on Rebuttal Revision**

Dear Area Chair and Reviewers,

Thank you for raising these concerns and questions and we have thoroughly responded to them by carefully revising our manuscript. Specifically, we have significantly improved our manuscript from 19 pages to 26 pages, incorporating **more related works**, enhancing **motivation of utilizing LLM**, and supplementing **experiments**. In addition, minor issues are also corrected to ensure as clear as possible. The modifications to previous version are highlighted in **orange** (subtitle of the subsection is highlighted in **orange** if newly added). Overall changes are briefly summarized as below.


**1) More description of Alg.1, including the coherence of TS-RAG and ReinLLM (Reviewer eiSN, uNhX), and the retrieval dataset selection (Reviewer eiSN, uNhX).** We have added a new subsection **The architecture of ZeroTS** to show the data flow of our whole ZeroTS, and linked this part with Alg. 1. This subsection can answer the following two questions 1) **Coherence between two model** and 2) **Dataset selection for retrieval (Reviewer eiSN, uNhX)**.

**2) Clarification of Information Leakage Risks (Reviewer YKdy).** We have added a new subsection for such clarification as A.1.4.

**3) Discussion on dataset selection. (Reviewer uNhX).** We have added a new new subsection for such clarification as A.1.5.

**4) More explanation of HNSW.** We provide more explanation of HNSW and the computation tool of Faiss. We have especially added three references (Johnson et al.(2019), Malkov & Yashunin (2018), Zhong (2020)) in Section 4.1. We also added more details in A.1.6 in our Appendix describing HSNSW, where modified sentences are **highlighted in orange**.


**5) More detailed discussions among ZeroTS and other related works.** The reviewers suggest comparing ZeroTS with other non-LLM based foundational models, with RLHF, with recent LLM such as Llama-3 and other cross-domain learning tasks. To respond, we have provided a systematical discussion among ZeroTS and other related works and added a new subsection entitled ‘**Distinguish ZeroTS against other related works**’ as A.2. It includes three subsections corresponding to following three questions:

-	Discussions between ZeroTS and RLHF (**Reviewer fTsg**);
-	Discussions on motivation of LLM and the version choice of LLM (Why LLM is selected and the version choice of GPT-2) (**Reviewer eiSN, 11xC, YKdy**);
-	Comparison with fine-tuning models (W2, **Reviewer 11xC**).

**6) Discussion on future work.** We have discussed three lines of future works in Sec A.5, 1) **Efficient retrieval strategy (Reviewer eiSN)**, 2) **Potential improvement on long-short term transfer (Reviewer eiSN)**, 3) **Practical deployment of ZeroTS with TD-engine (Reviewer 11xC)**.

**7) New results and hyperparameter analysis.** Based on advice of reviewers for enhancing the accuracy and comprehension, we have added experiments regarding above **five aspects**,

-	Horizon-level results (Table 6 and 7 in Appendix, **Reviewer fTsg**);

-	Ablation on Recent Llama-3 and PatchTST (Table 8 of Appendix, **Reviewer YKdy**);

-	Out-of-the-box zero-shot tests (Table 9 of Appendix, **Reviewer YKdy**);

-	Significance tests (Table 10 and 11 in Appendix, **Reviewer 11xC**);

-	A broader hyperparameter analysis (A.4.7 of Appendix, **Reviewer eiSN**);

-    Effectiveness of similar meta information selection  (**Reviewer uNhX**);

-  Impact of auxiliary dataset size (**Reviewer uNhX**).

We added **additional experiments** on **out-of-the-box zero-shot** and report the results for comparison.


**8) Motivation discussions.** In A.2, we have added a new section entitled  *Distinguish ZeroTS against other related works* to disentangle our work against others and provide more evidence for our motivation.


 **9) Some minor issues on expressions and formats.**

-	We have revised ‘time series prediction’ into ‘time series forecasting’ and abbreviate it as ‘TSF’ in title, Abstract and Definition. And then we modified the color of references into blue  and utilized ‘\citep{}’ for references (**Reviewer YKdy, uNhX**);

-	We have removed the subcaption in Fig.1 into the textual subcaption below the figure (**Reviewer uNhX**);

-	We have changed ‘distance’ in Eq(2) into ‘similarity’ for more accurate expression (**Reviewer fTsg**);

-	We have supplemented how we utilize meta information in both retrieval and ReinLLM (**Reviewer fTsg**, Appendix A.1.1);

-	Confusion of the contribution in constructing a retrieval database. We have modified ‘database’ into ‘base’ or ‘dataset’, and notated the ‘dataset’ besides ‘database’ to avoid the misunderstanding of constructing the database system (**Reviewer 11xC**);

-	**All the related works suggested by the reviewers** have been **incorporated** into our  revised version.

Thanks again for all above insightful suggestions, and we are looking forward to interact with you in more discussions in the following stages.


Best,


Authors of Paper 705.

---

### Author Response · Authors · 2024-11-27
**Thank you for your great efforts on Paper 705**

Dear AC and reviewers,

The authors appreciate the efforts made by AC and reviewers on our manuscript. We wonder whether our responses and revisions have well  addressed your concerns. If you have any further questions, we are  always glad to provide more discussions and explanations. Thank you.

Best,


Authors of Paper 705

---

### Note · Authors · 2025-01-22

I have read and agree with the venue's withdrawal policy on behalf of myself and my co-authors.